# Temperature and precipitation affect the water fetching time burden in Sub-Saharan Africa

Abigail Harvey Paulos[1], David A. Carroll II [2], Julie Powers[1], Jake Campolo[3], Daehyun Daniel Kim [1], Avery Cohn[2] & Amy J. Pickering [1,4,5] ✉

In Sub-Saharan Africa (SSA), over 75% of households lack on-premises water access, requiring residents to spend time walking to collect water from outside their homes – a time burden that falls disproportionately on women and girls. Climate change is predicted to alter precipitation and temperature patterns in SSA, which could impact household water access. Here, we use spatial first differences to assess the causal effects of weather on water fetching walk time using household survey data (n = 979,759 observations from 31 countries) merged with geo- and temporally-linked precipitation and temperature data over time periods ranging from 7 to 365 days. We find increases in precipitation reduce water fetching times; a 1 cm increase in weekly rainfall over the past year decreases walking time by 3.5 min. Higher temperatures increase walk times, with a 1°C increase in temperature over the past year increasing walking time by 0.76 min. Rural household water fetching times are more impacted by recent weather compared to urban households; however, electricity access in rural communities mitigates the effect. Our findings suggest that future climate change will increase the water fetching burden in SSA, but that co-provision of electricity and water infrastructure may be able to alleviate this burden.

Globally 1.8 billion people fetch drinking water from sources located off-premises[1]. In Sub-Saharan Africa (SSA), 76% of the population lack on-premises water access[2], with an estimated 20% having to walk over 30 min to their primary water source[3–6]. The burden of water collection associated with the lack of on-premises water access contributes to gender inequalities in SSA, as the responsibility for water collection falls primarily on women and girls[4,5]. Girls' school attendance has been found to decrease with time spent on household chores, including water fetching[7,8]. Previous evidence from SSA also indicates longer walk times between the home and water source can negatively impact child health, as a 15-minute reduction in one-way walk time has been associated with a 41% reduction in diarrhea prevalence and an 11% reduction in under-five mortality[9]. This effect could be driven by longer walking times reducing the total quantity of water collected by households and subsequently reducing water usage for hygiene and cleaning purposes[9–13], growing crops, and other income-generating activities. Long walk times can also necessitate lengthier storage time of water in the household, resulting in microbial contamination[14–18]. Together, these mechanisms increase the risk of child diarrhea and mortality.

Provision of safely managed drinking water can be limited by lack of electricity access. In Africa, over 40% of people lack access to electricity[19]. In Nigeria, a household survey found that electricity significantly influenced access to improved water sources: 93% of households with access to piped water had electricity access, and public tap usage was greater in households with electricity (76%) than

[1]Department of Civil and Environmental Engineering, University of California, Berkeley, USA. [2]Friedman School of Nutrition Science and Policy, Tufts University, Boston, MA, USA. [3]Farmer's Business Network, San Carlos, CA, USA. [4]Chan Zuckerberg Biohub, San Francisco, CA, USA. [5]Blum Center for Developing Economies, University of California, Berkeley, Berkeley, CA, USA. ✉e-mail: pickering@berkeley.edu

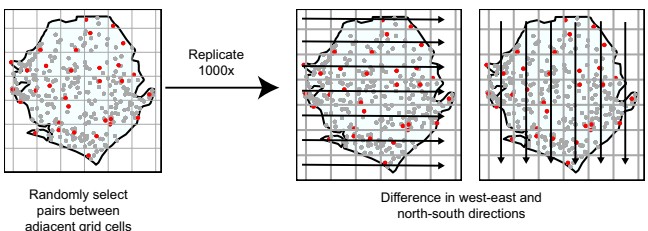

**Fig. 1 | Schematic of the spatial first differences (SFD) method using example data from Sierra Leone.** Red points represent random selection of households per grid cell. Grid cells represent the scale of the gridded precipitation data, 0.5° scale. The analysis is simulated 1000 times. SFD models are estimated west-east and north-south as a robustness check.

those without (21%), as was borehole usage (66% with access *versus* 34% without access)[20]. A separate study utilizing panel data collected between 2000 and 2020 in Africa found that electricity access was associated with increased access to safely managed drinking water[21]. Electricity access has yet to be directly linked with water proximity, yet the provision of water via improved sources often necessitates electricity access at least at the community-level to enable extracting groundwater via boreholes and transporting water to households through piped systems. Throughout SSA, both electric and manual pumps are common for extracting groundwater, suggesting that future increases in electricity provision could enable more reliable access to improved water sources[22,23].

Climate change has been highlighted as a particular risk to water access in SSA. Water availability and water fetching times are dependent on freshwater availability, which is known to vary seasonally with precipitation and temperature[9,12,24–27]. Climate change is predicted to increase annual average temperatures across SSA by as much as 4 °C by 2100, cause decreases to annual precipitation by up to 20% in some areas, and increase the frequency of extreme weather events such as droughts, floods, and heat waves[28]. Further, climatic variability, e.g. short-term weather, has been linked with elevated child diarrhea incidence in Africa, suggesting climate change will also have impacts on child health[29–31]. One mechanism by which climate change may impact child diarrhea is through water fetching, as longer water fetching times are associated with elevated child diarrhea[9]. However, the impact of weather on the water fetching burden in SSA has yet to be examined.

The impacts of weather on water fetching walk time could differ across various time scales. For example, in the very short term (7 days or less), households may switch sources immediately following heavy rainfall when they view certain source types as contaminated[27]. Conversely, heavier rainfall in the prior months may enable switching to surface waters or shallow wells as availability increases[32]. Periods of extensive heat and limited rainfall may result in sources drying up necessitating source switching, and longer-term weather may impact availability of groundwater sources[12,33]. Economic factors are also expected to play a role in household source selection; consumers may prefer surface water or shallow wells over those that charge collection fees when those source types are available[32]. Seasonal to annual weather can also impact crop yields and agricultural outputs, which in turn can affect households' ability to purchase water[34,35].

In this work, we quantify the impact of preceding weather on water collection walking time using almost 1 million geo-located Demographic and Health Surveys (DHS) linked to local temperature and precipitation data preceding each household survey date. We use spatial first differences (SFD) to estimate the causal relationship between walk time and preceding temperature and precipitation levels; SFD is a causal inference technique similar to first differencing approaches in time-series models, with the time index replaced by a space index (Fig. 1). Our main objective is to determine the effect of

recent precipitation and temperature levels on water collection walk time, and evaluate if community electricity access mitigates this relationship. In addition, we investigate if preceding weather is associated with the type of water source households use. Finally, we explore whether urbanicity, climate region (arid, temperate, or tropical), and access to improved water infrastructure influence how weather affects the walk time burden.

## Results
The combined household and weather dataset included 104 surveys conducted between 1990–2017 in 31 countries in Sub-Saharan Africa, with $n = 979,759$ water-fetching observations. Data availability, walk time to water source, daily precipitation, and daily maximum temperature varied by country and region (Fig. 2; Fig. 3; Supplementary Table 1; Supplementary Figs. 1, 2, and 3). Overall, 80% (SE: 0.04%) of households reported collecting water outside the home and 41% (SE: 0.02%) reported a walk time exceeding 30 min round-trip. Approximately two-thirds of data are in rural regions ($n = 663,290$ rural *versus* $n = 322,353$ urban). Mean walk times were greater and more variable in rural households (*t*-test, $p < 0.001$; F test for variances, $p < 0.001$); the median walk time in rural households was 15 min (SE: 39.0) *versus* 5 min (26.3) in urban households. The household member responsible for water collection was only reported for 15% of observations ($n = 150,541$); of these, women were responsible 67% of observations, girls 10%, men 14%, and boys 4%, with 5% of households reporting 'other'; these categories were used as defined by the DHS where adults are aged 15 years and older. See Supplementary Note 1 for additional household characteristics.

### Weather affects water fetching walk times
Using SFD, we estimated the causal relationship between walk time and preceding temperature and precipitation levels, including over the past 7 days, 30 days, 90 days, 180 days, and 365[36]. The SFD method assumes exposure (temperature and precipitation) and unobserved confounding variables are not systematically correlated between adjacent clusters. SFD was selected over other causal inference techniques due to its ability to account for unobserved heterogeneity (confounders). The DHS program reports a Global Positioning System (GPS) coordinate for clusters of 25–30 households located in the same geographic area. Households were linked to weather data based on the date of the survey and the grid cell their cluster was located in. We performed simulations to randomly select pairs of households from adjacent grid cells, difference a household's walk time and weather data across adjacent cells, and estimate the effect of weather over various time periods on household walk time with regression models. We estimated SFD west-east as the primary analysis, and north-south as a robustness check. This analysis was simulated 1000 times and aggregate estimates analyzed. We also conducted stratified analyses by rural *versus* urban populations, and in rural households by those *with versus* without access to electricity (within rural households only since electricity variation was limited in urban settings).

We find that precipitation and temperature affect the one-way walking time to a household's primary drinking water source. Across all time periods, precipitation is inversely associated with water fetching time, and the magnitude of effect increases with the length of the time period. In the 7-day time period, a 1 cm increase in daily precipitation is associated with a 0.2 min decrease in walk time, while a 1 cm increase in daily precipitation in the 365-day time period is associated with a 3.5 min decrease in walk time (Fig. 4a). Daily maximum temperature is positively associated with walk time across all time periods. The magnitude of effect also increases with the length of the time period; a 1 °C increase in mean daily maximum temperature over the last 7 or 365 days is associated with a 0.47 or 0.76 min increase in walk time, respectively. Comparing the effects across weather variables, an increase of one standard deviation (σ) in the 365-day time period in

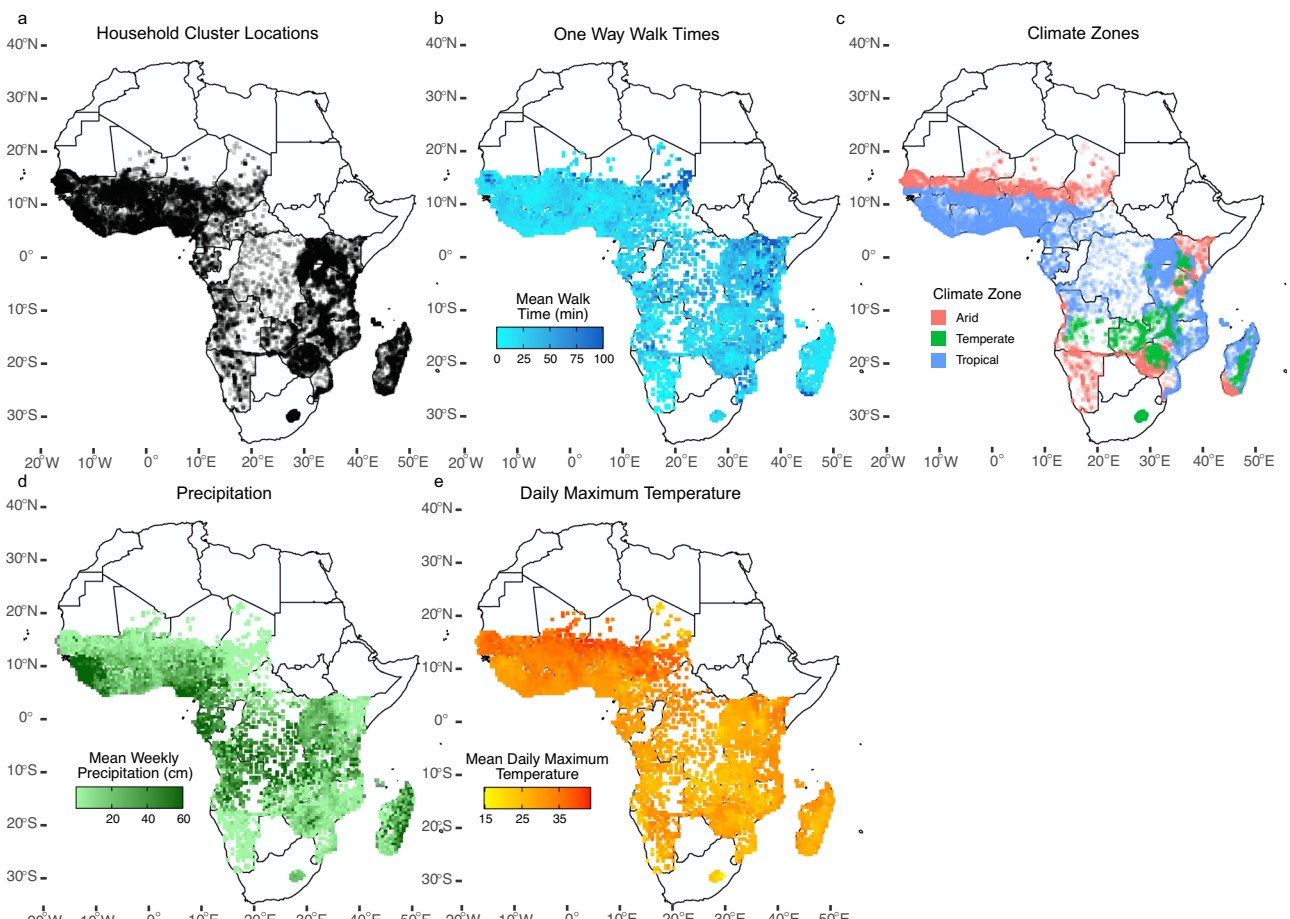

**Fig. 2 | Household survey locations and mean daily weather over the year leading up to the survey, aggregated by weather grid cell (0.5° resolution).**
**a** Locations of household clusters as black dots of survey data used in this paper collected between 1990 and 2017. **b** Mean one-way walk time, with darker colors indicating a longer mean walk time. **c** Location of household clusters, colored by climate zone categorization. **d** Mean daily precipitation, averaged over the 365 days leading up to the survey. **e** Mean daily maximum temperature, averaged over the 365 days leading up to the survey.

precipitation is associated with a greater magnitude of impact on walking time (−4.34 min) than an increase of one σ in daily maximum temperature (3.04 min).

We conducted the SFD differencing in two directions, north-south and west-east (Supplementary Note 2; Supplementary Fig. 4; Supplementary Tables 2–6). We also conducted additional SFD sensitivity analyses, including SFD models with temperature and precipitation binned by the 75th and 90th percentiles (Supplementary Tables 7, 8), SFD models using average weather and walk time per grid cell (Supplementary Table 9), and SFD models with data points within 5 km (rural) or 2 km (urban) of grid cell borders removed to account for DHS GPS coordinate jittering (Supplementary Table 10). While magnitude of effect varied, direction of effect broadly agreed across all models (Supplementary Fig. 4).

As another comparison, we also conducted fixed effects models controlling for year, month, country, and county/subnational region. The fixed effects models agreed with SFD models on direction of effect, with usually a larger magnitude of effect (Supplementary Note 3; Supplementary Fig. 5). In the fixed effects models, we find that urban/rural status impacts the effects of weather on water fetching walk time. Urban households are less impacted by precipitation and more impacted by temperature than rural households (Supplementary Tables 11–20). These findings suggest that urban areas may be less susceptible to precipitation impacts than rural households. However, the fixed effects results cannot be interpreted as causal as they may be impacted by unobserved confounders.

Urban households were excluded from subgroup analysis in SFD models, as urban grid cells had sparse neighboring grid data (only 66% of available data could be included in SFD models) due to lack of data in adjacent grid cells. The grid cells for our weather data are 0.5 degrees, approximately 55 square kilometers at the equator, which can be larger than an urban city or town area. Further investigation is needed to determine the impacts of climate on urban water fetching.

## Electricity mitigates weather impacts in rural areas

In SFD analyses stratified by electricity access, we found the effects of precipitation on water fetching time were greater in rural households without electricity than those with electricity access (−5.9 *versus* 0.03 min/cm for precipitation in the 365-day time period; Fig. 4). Effects of temperature on walk time did not vary by electricity access. Among rural households, 11% had access to electricity. Water source use differed significantly by household electricity access: improved sources were used by 64% of households with electricity *versus* 51% of those without and piped water was used by 45% *versus* 19%. Conversely, households without electricity more commonly used boreholes (19% *versus* 12%), shallow wells (20% *versus* 11%), and surface water (35% *versus* 29%). All differences were statistically significant by t-tests ($p < 0.05$).

We hypothesized that wealth and electricity access may be closely related in our dataset, such that wealth would be systematically higher in households with electricity than in those without. To examine the

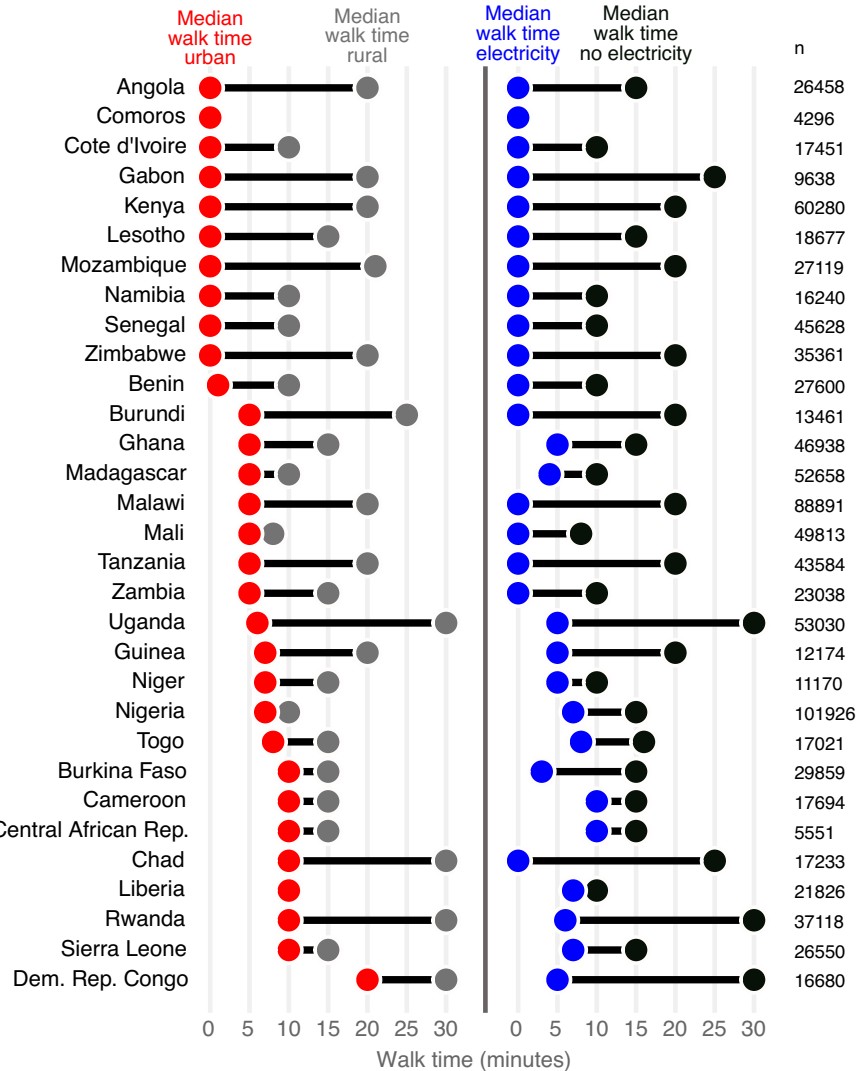

**Fig. 3 | Median walk time in minutes by country, disaggregated by rural *versus* urban status and electricity access.** Black lines are a visual aid only.

association between wealth and electricity access, we converted the wealth scores reported within DHS surveys to a comparable scale using the method of Rutstein & Staveteig[37]. Overall, households with electricity access had significantly higher household wealth than those without in rural households and in the overall dataset (*t*-tests, $p < 0.001$; Supplementary Fig. 6). In rural households, fewer than 5% of households with electricity had wealth scores below the median. In fixed effect models adjusted for household wealth (Supplementary Tables 11–20), we found the same direction of effect as in the unadjusted models but with attenuated point estimates. When electricity access is considered on the community scale (Supplementary Methods), effect estimates are similar to those for electricity access at the household level (Supplementary Table 21).

**Effects differ by climate regions**

We explored if weather effects on walking time differed by arid, temperate, or tropical climate regions. We categorized all data points by Koppen–Geiger climate classification for the time period between 1980–2016[38] and binned data by arid, temperate, or tropical climate zone. Stratified SFD analyses were conducted by climate zone. Twenty percent of data points fell in arid regions, 17% in temperate regions, and 63% in tropical regions (Fig. 2). As expected, precipitation is greatest in tropical zones and least in arid zones. Temperature is greatest in arid regions, followed by tropical

regions, with the lowest mean temperatures in temperate zones (Supplementary Table 22).

For temperature, effects vary by time period (Fig. 4b). In the 7-day time period, arid regions are less impacted by temperature increases than tropical or temperate regions, and in the 30- and 90-day time period, tropical regions are more impacted than arid or temperate regions. For the 90-day time period, arid regions are inversely associated with temperature (a 1C increase in temperature is associated with a 0.13, 95% CI: 0.01–0.25 min decrease in walk time), and in the 180-day lag, temperate regions are inversely associated with increased temperature (a 1C increase in temperature is associated with an 0.12, 95% CI: 0.03–0.21 min decrease in walk time). In the 365-day time period, temperate regions are more impacted by temperature increases than tropical regions, while arid regions are the most impacted by temperature but with larger uncertainty in the estimate of effect.

**Effects differ by improved water source usage**

To examine whether the use of improved source types resulted in increased resilience to changes in short-term weather, we conducted SFD analyses stratified by access to improved drinking water infrastructure. Improved infrastructure was defined as the following reported primary drinking water source types: borewells/boreholes, piped water, rainwater, protected springs, and protected wells. We find

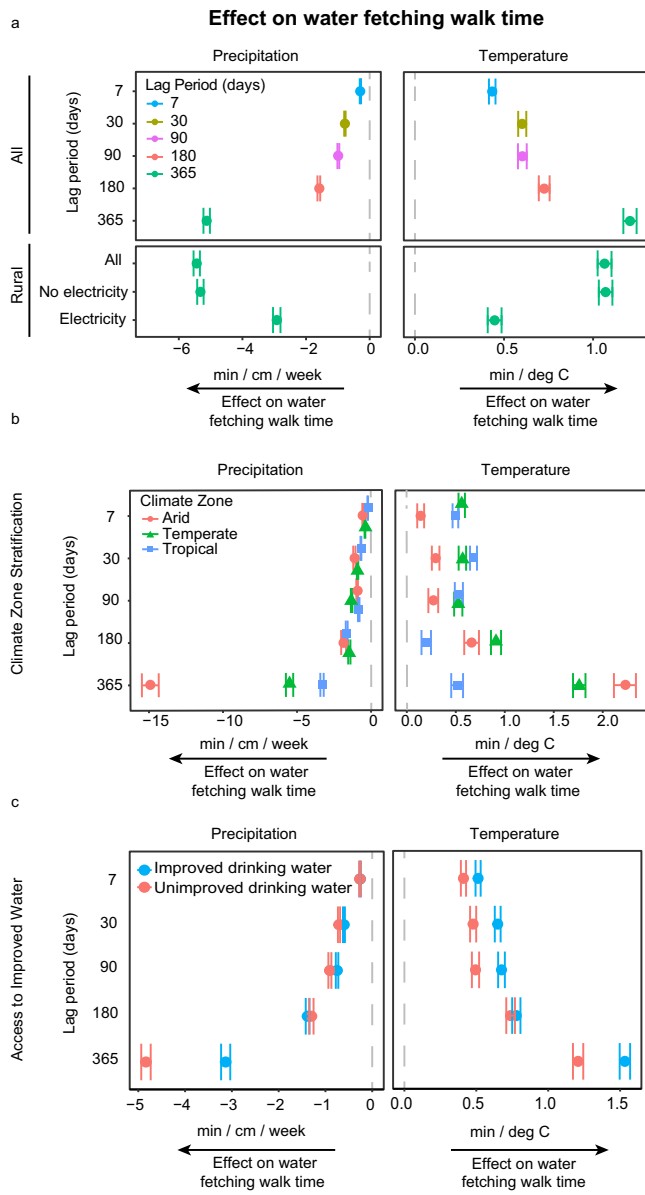

**Fig. 4 | Effect of recent changes in precipitation and temperature on water fetching walk time as estimated by spatial first differences (SFD). a** Effect estimates (regression coefficient on walk time) for 7, 30, 90, 180, and 365 day time periods among all households in top box, and effect estimates among rural households only in the bottom box for the 365-day time period, stratified by electricity access. **b** Effect estimates stratified by climate zone. **c** Effect estimates stratified by access to improved *versus* unimproved drinking water. SFD effect estimates are for precipitation time variables (left; minute/centimeter/week) and temperature time variables (right; minute/degree). For all panels, the circles represent the mean effect estimate and error bars represent the 95% confidence intervals around the point estimate from $n = 1000$ model replicates. Source data are provided as a Source Data file.

that households with access to improved sources are less impacted by precipitation and temperature across all time periods; in the 365 day time period, households with access to improved sources are less impacted by precipitation and temperature by an approximate factor of 2 (Fig. 4c).

### Weather affects water source type usage
One mechanism by which weather can impact water fetching time is through water source switching. Different weather conditions may impact source availability and consumer preferences, perhaps

necessitating use of a more distant water source. To assess the impact of weather variation on source utilization, we conducted logistic regressions with source type usage as a binary dependent variable and precipitation and temperature as independent variables in separate models. In urban areas, elevated temperatures are associated with increased odds of borehole and well source use and decreased odds of surface water use. In rural areas, elevated temperatures are associated with increased odds of borehole and well source use and decreased odds of piped water use (Fig. 5). Associations between precipitation and source type use are more limited. In urban areas, increased precipitation is associated with decreased odds of piped water usage in time periods 90 days or shorter, and increased odds of well water use over all time periods. For rural areas, increased precipitation is associated with decreased odds of use of boreholes.

## Discussion
Here we show that the time burden of water fetching in SSA, and particularly in rural SSA, is susceptible to changes in precipitation and temperature. Our findings imply that future climate change will increase the water fetching burden in SSA. On average, if a household takes 4 trips to collect water per day[39], a 4 °C increase in daily maximum temperature[40] would be equivalent to an 85 min increase in total walk time per household per week. The same 4 °C temperature change would increase the percentage of people walking >30-min round-trip from 41% to 50%. A 10% decrease in precipitation[28] would be associated with an increase in weekly walk time of 22 min. A universal 10% decrease in precipitation (although not likely) would change the fraction of people walking more than 30-min to their water source from 41% to 67%. The water fetching burden is more affected by changes in precipitation than temperature on a per σ basis; however, variability in weather is expected to change (and increase) with climate change which may impact the relative importance of these weather variables on water fetching[41]. The majority of the increased water fetching burden under climate change will fall on women and girls; in this study, females were responsible for water collection in 77% of households. Taken together with previous evidence, our results also suggest future climate change will increase the risk of child diarrhea and mortality through longer water fetching times in SSA[9].

One mechanism by which weather can impact water fetching time burden is through source switching. While the results of our source usage logistic models cannot be interpreted as causal, households often use multiple drinking water sources[33], and our results suggest that a household's choice of primary drinking water source is associated with seasonal weather. Hotter weather was associated with increased usage of boreholes and shallow wells and decreased usage of piped water and surface water. These results are consistent with previously published research in rural regions finding that surface waters dry up seasonally or become nonfunctional during periods of high temperatures and low rainfall[12,24,27,33,42]. Increased usage of borewells and shallow wells, but less usage of piped water supplies, could be explained by pumps moving water through piped systems being more likely to be powered by grid electricity (which is more intermittent in hot and dry periods), while water extracted from borewells is more likely to be powered by solar power or manual hand pumps. Increased precipitation was associated with decreased borehole usage in rural households, and increased use of shallow wells in urban areas. This differential response to rainfall in urban versus rural areas could be driven by differences in water infrastructure in rural *versus* urban areas. Taken together with our finding that increased precipitation reduces water fetching time for households, households are likely choosing to access closer sources, such as surface water in rural areas and shallow wells in urban areas, that become available during rainy periods.

We hypothesize that electricity may increase climate resiliency by enabling the movement of water[43]. The effects of precipitation on the

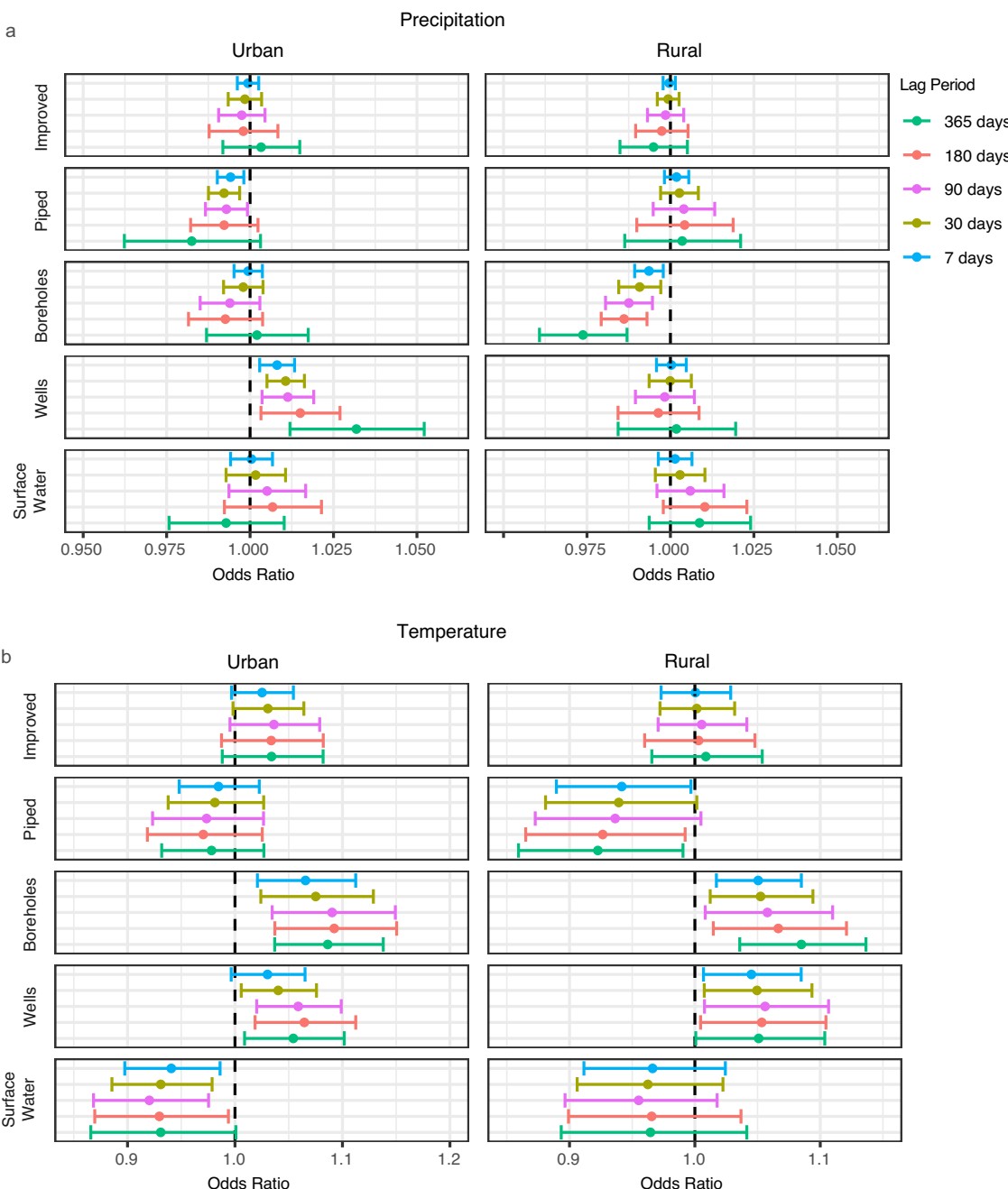

**Fig. 5 | Effects of recent weather on water source type use in rural *versus* urban regions by lag period. a** Odds ratios of water source type usage for a 1-unit increase in precipitation (cm/week). **b** Odds ratios of water source type usage for a 1-unit increase in temperature (degree). Logistic regression models control for country, year, and month and are conducted for improved *versus* unimproved, and the following water source types (*versus* all other sources): boreholes, piped water, shallow wells, and surface water. Circles represent the point estimate of the model, while error bars represent the 95% confidence intervals. Source data are provided as a Source Data file.

water fetching burden were lessened for households with electricity access in their home, and households with electricity access relied less on boreholes and shallow wells and more on piped water than those without electricity. Electricity can power pumps to transport water into communities and households and to fill storage tanks, perhaps increasing resiliency to short-term changes in climate. Electricity availability in one's community, even when not all community members have a household grid connection, may still enable this transport of water and may alleviate the impacts of climate on water fetching for all community members. We also find that wealth and electricity access were strongly related. Wealthy households are more likely to afford an electrical connection, but electricity access can also contribute to

wealth-generating activities. Electrical power can be used to reduce time spent on household chores and increase time spent on income-generating activities[44,45]. Considering electricity access is intertwined with wealth, we were unable to fully differentiate between the impacts of electricity access and above-average wealth. Further research is needed to determine if there is a causal effect of electricity access mitigating the effect of weather on the water fetching burden.

The magnitude of effect of temperature and precipitation on water fetching walk time differed by climate zone, with arid and temperate regions most susceptible to weather changes as compared to tropical regions. Arid regions are known to be vulnerable to the impacts of climate change due to limited infrastructure, services, and

institutional capacity, poorly adapted crop varieties, livestock not coping with heat, and a reliance on rain-fed crops[46]. Tropical regions have ecosystems that are among the most resilient to climate change of all ecosystems[47], which may increase the resiliency of water availability to changes in weather.

Long-term changes to climate are likely to affect water availability and water fetching burden in SSA differently, and perhaps more significantly, than is observed through short term variation in temperature and precipitation. A limitation of this study is that we were only able to quantify the effects of weather (climatic variability) on water fetching rather than the effects of climate change. While techniques utilizing data on climatic variability to understand the potential impacts of climate have been used previously[48–54], the ideal scenario would be to use long-term climate data directly. However, data on water fetching and weather in SSA were sparse or nonexistent pre-1990s, leaving little to no historical data to examine. Further, the entire continent (and world) is undergoing climate change, and there is no valid counterfactual to determine water fetching in lieu of climate change. While outside the scope of this paper, our results could be used to inform modeling using various Representative Concentration Pathway (RCP) scenarios to predict the water fetching burden in SSA under future climate change. Further, prior research has found that short-term weather impacts child diarrheal disease incidence[29–31]; future work is needed to determine how much of this effect is mediated through walk time.

Broadly, our results highlight the importance of climate-resilient water provision in Sub-Saharan Africa. Weather events driven by climate change threaten progress in SSA towards universal access to reliable on-plot clean drinking water outlined in Sustainable Development Goal 6. We found consistent inequalities in the impacts of climate change on water fetching, which is useful for identifying priority investment areas. Rural households, especially those without electricity access, are particularly vulnerable to the effects of weather on water fetching time. We also found that effects vary by climate zones, with those living in arid and temperate regions most vulnerable to changes in weather. Electricity access mitigated the effect of weather on water fetching time in rural areas, adding to the broad range of benefits from communities gaining access to electricity[55]. Our results suggest that diverse energy sources could improve climate resilience of water supplies, as solar power would be available during hot and dry periods, while hydropower is available during rainy periods[56]. Water delivery and energy infrastructure are often rolled out separately in rural African communities[57], yet our results suggest that governments may want to consider investments in integrated water-energy infrastructure to enhance climate resilience and benefit human health.

## Methods
### Data sources
We obtained household survey data from the Demographic and Health Survey (DHS) Program for countries in Sub-Saharan Africa. We utilized both DHS and Malaria Indicator Survey (MIS) datasets available for 31 countries from 104 DHS and MIS survey rounds conducted between 1990 and 2017. Our data set included 985,643 household surveys. The survey variables included date of survey, GPS coordinates for households at the cluster level (10–50 households), one-way walk time to water source (in minutes), primary drinking water source type, rural *versus* urban designations, household electricity access, and other variables relating to wealth. Water sources were categorized as improved or unimproved per JMP definitions. Improved sources included borewell, piped water, rainwater, protected springs, and protected wells. Unimproved sources included bottled water, vendor water, unprotected springs, surface water, and unprotected wells. 'Other' sources were uncategorized and treated as missing.

Daily temperature data at 0.5° resolution were obtained from the NOAA NWA Climate Prediction Center (CPC)[58]. Daily precipitation data

at 0.25° resolution were obtained from the Climate Hazards Group InfraRed Precipitation with Station Data (CHIRPS) global rainfall dataset[59]. Climate zone classifications were obtained from maps created by Beck et al.[38].

We merged household survey data with the CPC and CHIRPS meteorological data to create a combined dataset of spatiotemporal data of water fetching time and meteorological data. DHS and MIS data points were aligned spatially and temporally with CPC and CHIRPS meteorological data. For each household survey data point, we extracted daily precipitation and temperature in the survey location for 0-365 days leading up to the survey date. We created variables containing average daily maximum temperature for several time periods leading up to the survey date: 7 days, 30 days, 90 days, 180 days, and 365 days. Precipitation variables over the same time periods were created by summing daily precipitation over the prior 7, 30, 90, 180, and 365 days, and dividing by the number of weeks per time period to calculate cm/week of precipitation leading up to the survey date.

Data points with missing walk time to water source were excluded from the dataset. Data points with GPS coordinates falling outside their respective country by greater than 10 km were also excluded; country extents were determined from the UIA World Country Boundaries[60]. We also excluded data with a survey date that fell outside the survey data collection dates listed by DHS for a particular survey round, with a two-month buffer around start and end dates.

### Spatial first differences regression models
We estimated regressions using spatial first differences (SFD) (Fig. 1)[36]. The SFD method can identify causal effects with the assumption that changes in weather and unobservable confounders are not systematically correlated between adjacent grid cells[36]. We utilized the grid cells of the temperature and precipitation data to conduct the differencing. Differencing was restricted to within each DHS country survey to ensure data are temporally aligned. Household data collected through DHS/MIS are collected by cross-sectional surveys conducted during a time span of several months to 1 year, with a median collection time of approximately 5 months. When we create our pairs by differencing, the mean difference between survey collection dates is approximately 30 days.

Because SFD eliminates bias due to spatially correlated variables, we opted not to include covariates in our main models that could be along the causal pathway of weather effects on water fetching time (i.e., wealth, water infrastructure, freshwater availability).

First, we subset data to one of the 104 country surveys and then randomly selected pairs of data from adjacent grid cells. The maximum number of available pairs that avoided repeating observations was used for each grid-cell pairing. We differenced walk time and weather values of these randomly selected pairs in adjacent grid cells, moving in the west-east direction. We executed the differencing separately for each of the 104 country surveys and merged the results into one dataset. Then, we conducted a linear regression as Eq. (1):

$$WT_i = \beta_0 + \beta_1 * WV_{k,i} \qquad (1)$$

where WT is the one-way walk time to water source and $WV_k$ is the $k^{th}$ weather variable, including averaged daily precipitation and daily maximum temperature across all time periods. We replicated this analysis 1000 times and aggregated results. The standard deviations of our estimates were assumed to be equivalent to the standard error and were used directly in calculating confidence intervals[61]. As a robustness check, we also performed this analysis by differencing in the north-south direction.

We repeated this analysis within rural households, and within rural households we stratified by those with *versus* without electricity. To examine effects by climate zone, we repeated this analysis stratified by the three climate zones. Variation of effects by access to improved

drinking water sources was examined by stratifying by access to improved drinking water.

## Sensitivity analyses

By selecting the maximum number of pairings between grid cells and conducting 1000 replicates, we leverage the full variability of the dataset. However, not all households are included in each iteration of the model. To examine the effects of this data loss on the model outputs, we also estimated models using average weather and walk time per grid cell (see Supplementary Table 22) such that all data points were included in one model.

We conducted the following sensitivity analyses: comparing East-West and North-South SFD models (Supplementary Fig. 3), SFD models with temperature and precipitation binned by the 75$^{th}$ and 90$^{th}$ percentiles (Supplementary Tables 19, 20), SFD models with data points within 5 km (rural) or 2 km (urban) of grid cell borders removed to account for DHS GPS coordinate jittering (Supplementary Table 21) and fixed effects regressions (Supplementary Tables 9–18; Supplementary Fig. 4).

## Fixed effects regression models

We performed fixed effects regressions as a secondary analysis to compare to the spatial first differences method. Using the same combined dataset as in the spatial first differences approach, fixed effects regressions were estimated at the household level for each climate variable and time period combination, including fixed effects for county/subnational district, country, year, and month as Eq. (2):

$$\mathrm{WT}_i = \beta_0 + \beta_1 \mathrm{WV}_{k,i} + \beta_2 \mathrm{Country}_i + \beta_3 \mathrm{County}_i + \beta_4 \mathrm{Year}_i + \beta_5 \mathrm{Month}_i \tag{2}$$

where WT is the one-way walk time to water source and WV$_k$ is the k$^{th}$ weather variable, including daily precipitation and daily maximum temperature across all time periods. We calculated robust standard errors for heteroskedasticity clustered at the DHS household clusters.

To determine the impacts of rural/urban and electricity status on the relationship between weather and walk time, we included an interaction term in the models (see Supplementary methods and supplementary equations (1) and (2)).

## Water source use regression models

To assess the impacts of weather on drinking water source type use in our dataset, we used logistic regression, controlling for country, year, and month, and clustering standard errors by DHS household clusters as Eq. (3):

$$\mathrm{Source}_{k,i} = \beta_0 + \beta_1 \mathrm{WV}_{k,i} + \beta_2 \mathrm{Country}_i + \beta_3 \mathrm{Year}_i + \beta_3 \mathrm{Month}_i \tag{3}$$

where Source is a binary variable indicating k$^{th}$ type of source (e.g. improved versus unimproved, piped versus all other sources, etc.) used by household i and all other variables as described above. Results are presented as odds ratios.

Analyses were conducted in R version 3.3.

## Reporting summary

Further information on research design is available in the Nature Portfolio Reporting Summary linked to this article.

## Data availability

The water fetching and climate data are available in a public repository at https://github.com/abharv52/Water_fetching_climate Source data are provided with this paper.

## Code availability

All code is available at https://github.com/abharv52/Water_fetching_climate[62].

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

## Acknowledgements

We thank Andre Dozier and Mariella Medina Castellanos for research assistance. This research was funded by the Tufts University Collaborates Seed Grant Program to A.J.P. and A.C., the Chan Zuckerberg San Francisco Biohub Investigator to A.J.P., and the National Science Foundation to A.H.P. (DGE-2125913).

## Author contributions

A.J.P. and A.C. conceived of the study. A.H.P., J.C., J.P., and D.A.C. curated the data and conducted the analyses; A.H.P. generated tables and figures. D.D.K. replicated analyses. A.H.P. and A.J.P. wrote the first draft. A.H.P., A.J.P., A.C., J.C., J.P., D.A.C., and D.D.K. edited the final draft.

## Competing interests

The authors declare no competing interests.
