## [Transparent Peer Review file · Nature Communications]

Temperature and precipitation affect the water fetching time burden in sub-Saharan Africa

Corresponding Author: Dr Amy Pickering

Version 0:

Reviewer comments:

Reviewer #1

(Remarks to the Author)

Reviewer's Comments on "Temperature and precipitation affect the water fetching time burden in sub-Saharan Africa".

General comment: The main objective of this study is to assess the impact of weather variability, as measured by temperature and precipitation data, on the time required to travel to gather drinking water in SSA. The research topic is contemporary and pertinent, considering the increasing weather variability and the limited coping capacity of most households in SSA. The paper has the potential to contribute to the field of climate change and vulnerability by examining one of the dimensions of household vulnerability to the effects of climate change. In the reviewer's opinion, the paper nevertheless needs to address the following key issues:

Major comments:

One of the paper's objectives is to "evaluate if community electricity access mitigates the relationship between weather and the water fetching time burden." However, given that access to electricity is extremely limited in sub-Saharan Africa, particularly in rural areas, the authors did not provide a compelling reason for this. This was not explicitly motivated in the main discussion section; it simply came as a surprise that they will conduct the analysis based on access to electricity.

The authors argued that access to electricity allows households to pump water from boreholes. However, this is especially true if an electric pump is the primary pump used. Is this true in most SSA countries? If this is the case, please give some supporting data/literature. If manual borehole pump is the most widely utilised technique, the case for using electricity as the primary means of mitigation the impact of weather variability on walking time to fetch water is not compelling.

The paper lacks a coherent conceptual framework explaining how short-term weather variability is predicted to increase water fetching time in SSA. What are the primary pathways by which weather variability leads to increased water fetching time in SSA countries? Is this because of variations in the extent of improved water infrastructure development? If yes, did you control for such infrastructure development in your regression analysis? Are your results robust when you account for the proportion of households that have access to improved water sources, such as borewells?

What are the policy implications and priority areas based on this study? Improve access to electricity or water infrastructure to lessen the amount of walking required to fetch water. Your policy recommendation makes no mention of the role of water infrastructure development.

It is crucial to investigate if access to improved water infrastructure can alleviate the effects of weather variability on water fetching time. Are areas with access to improved water infrastructure better equipped to handle the impact of weather variability compared to those without such infrastructure? To do this, you can include an interaction term of water infrastructure development indicators with indicators of weather variability.

In relation to the prior comment, when detailing the regression approach, the authors simply indicated that they accounted for the availability of electricity and other wealth-related factors. What are the wealth-related variables you have included in your model?

The regression procedure and the variables used should be clearly stated in the main text. For example, the author said that "First, we subset data to one of the 104 country-surveys and randomly selected one data point in each grid cell." Please explain what you mean when you say you selected one data point per grid cell. Is it one data point for the weather data, or did you choose one data point from the sample households? Then, for your dependent variable, you use the average walking time to fetch water across all sample households related to each grid.

With regards to the regression of water source type utilization, the authors argued that weather variability can affect water fetching times through water source switching. However, the regression method used is not clear. Is the dependent variable change in the type of water source used (e.g., from piped to borewell?) and the independent variable change in temperature and precipitation? This should be explained clearly.

There is no convincing explanation provided for the results obtained when regressing water source type on weather data. For example, why do you expected to detect no impact of precipitation on the type of water source used, however you found a significant effect of temperature in rural locations.

Minor comments

The article needs proofreading.

In the abstract the author stated that

"In sub-Saharan Africa (SSA), over 75% of the population must spend time walking to collect water from outside their home, with women and girls bearing the majority of the time burden."

Is this a population, or a household? Not everyone in a household spends time collocating water. If so, it should be stated that "more than 75% of the population in SSA live in households where at least one member walks to collect water from outside their home."

(Remarks on code availability)

Reviewer #2

(Remarks to the Author)

This article empirically estimates the impact of temperature and precipitation on water fetching times in sub-Saharan Africa. Lack of access to safe drinking water is a widespread problem globally, and water collection imposes a significant burden on many of the world's poorest households. Understanding how this burden will be affected by climate change and identifying possible mitigation measures is an important area of research. However, while I believe the topic of the study is of sufficient general interest to merit publication in *Nature Communications*, I have major concerns around the design and execution of the study that call into question the reliability of the findings. Further, the paper reads as very unfinished to me – there are multiple interesting and relatively easy to implement (with their DHS data) additional analyses that would strengthen the causal claims made in the paper, shed more light on the mechanisms and consequences of increased water collection times, and increase the impact of the paper's results.

Technical issues

I have many technical concerns with the research design and implementation. One central issue is that the research design throws out a lot of useful variation (e.g. by only using a random subset of points in estimation and by only using spatial variation when temporal variation exists). A second major problem is that the specification of climate does not follow a large literature that finds that temperature and precipitation extremes and variability generally matter far more for the human impacts of climate than averages. Specific concerns are detailed below.

1. Why do the authors only select one data point per cell? This approach throws out a lot of useful information. There are at least two alternative strategies that would allow the authors to use all the data: (a) average water fetching times for all data points within a grid cell, or (b) for neighboring grid cells that each contain more than one observation, construct more than one differenced pair using independent data points. These approaches both seem far more standard and sensible to me than bootstrapping with 1000 repetitions.

2. The spatial first differences research design was designed for cross-sectional settings, in which no temporal variation is available (Druckenmiller & Hsiang, 2019). However, the authors do have temporal variation in both climate and water fetching times in their data. Why not use this variation to improve identification? While the DHS is not a longitudinal survey, it does provide a repeated cross-section, and the authors are already collapsing the data to a standardized grid. They could

therefore use temporal variation in their treatment and outcome variables collapsed to this grid. One simple way of doing so would be to calculate long differences (Burke & Emerick, 2016) between DHS surveys. Spatial first differences could then be applied to the long-differenced data. Alternatively, the authors could use a two-way fixed effects panel design.

3. It is not clear from the manuscript whether appropriate care was taken in the construction of the data to ensure the comparability of neighboring observations and the correct matching between DHS observations and climate variables.

(a) Are the pairs of neighboring observations that are differenced restricted to the same time period? The SFD approach removes all spatially correlated unobservable confounders; it does not remove temporally correlated confounders. Temporally correlated confounders likely matter a lot in this setting, as weather is highly correlated with time-varying human activities, such as time spent working in agriculture, that also affect water fetching times. Taking differences between data points recorded in different weeks would be problematic for the 7-day results. Taking differences between data points recorded in different years would be problematic for the 365-day results.

(b) DHS observations do not come with precise GSP coordinates – the coordinates are jittered up to 10km to protect privacy. This could cause an incorrect match between a DHS observation and the temperature and precipitation measures, which is important in the SFD context because the observation would be assigned the treatment of a neighboring grid cell. One easy fix is to drop all observations within 10km of the grid cell boundary in rural areas and within 2km of the grid cell boundary in urban areas (DHS jitter is 10km in rural areas, 2km in urban areas).

4. The authors specify the relationship between water fetching times and temperature and precipitation as linear. They conduct no robustness to the specification of this relationship. This choice is surprising given that the climate impacts literature overwhelmingly finds non-linear relationships between temperature and precipitation and human impacts (Hsiang, 2016). For example, counts of extreme heat days are far more predictive of climate damages than average temperatures, across a wide range of outcomes. And the quantity of rainfall that arrives in short, intense bursts often matters more than the average amount of precipitation. Furthermore, climate change is expected to impact climate extremes and variability just as much as averages. Best practices in climate econometrics suggest that the specification of temperature and precipitation should be non-linear. The authors could use bins or higher order polynomials and should test robustness to different specifications.

5. The fixed effects regression used for comparison does not have granular enough fixed effects to appropriately control for unobservable confounders. I would like to see a version of this equation with country-year-month fixed effects to account for common time trends within a country and subnational spatial fixed effects to control for time-invariant differences between locations.

6. The authors compare differences in treatment effects between subgroups using stratified samples. This does not allow for formal testing of these differences, such as whether climate matters more for households without access to electricity. The authors should pool the observations and include interaction terms to test for heterogeneous effects. In the SFD specification, the group variables used in the interactions should also be spatially first differenced.

7. The reported standard errors are extremely small. This may be because the authors do not account for spatial and temporal autocorrelation in the residuals.

8. The comparison of East-West and North-South SFD specifications is intended to test the robustness of the results to using different variation in data. While the estimates across these two directions consistently have the same sign, the magnitudes are quite different, falling well outside each other's 95% confidence intervals. This could reflect standard errors that are too small (see comment above), or it could be viewed as evidence that the identifying assumptions of SFD are not satisfied.

9. The Methods section is very thin and it is not always possible to follow what the authors have done. Much more detail is required. A few examples:

- It's hard to tell how the weather variables are constructed and defined. This is central to the interpretation of the results. For example, the authors write: "We created variables containing average daily weather for several time periods leading up to the survey date: 12 7 days, 30 days, 90 days, 180 days, and 365 days. The precipitation variables were divided by the number of weeks per time period, so that precipitation across all time periods are reported as cm of precipitation per week." Do you mean to say that you took the sum of daily precipitation over the time-period (rather than average) and then divided by the number of weeks? That would give you precipitation in cm/week, which is what I think you intend. But then in Table S1, median precipitation (measured in cm/week) is monotonically increasing in the lag period length. This suggests either that I have misinterpreted what this measure is or that there is an error in the data processing, as there is no obvious reason why average weekly precipitation should be longer for longer lag periods (unless all surveys are conducted in dry season).

- The estimating equations are not reported for many regressions other than the baseline specification. For example, the water source type use regressions (Figure 5), and the regressions that control for wealth are missing.

- The unit of observation in the fixed effect regression is never reported. Are we to assume that these regressions implemented on the same exact data frame as the SFD for comparability?

Additional analyses

10. The authors argue that expanding access to electricity should be viewed as a strategy for mitigating the negative impacts of climate on water collection times. This finding would have important implications for decision-making if it was well supported by the data. However, the analysis here is quite under-developed. It could be strengthened in at least two ways using DHS data:

(a) As the authors acknowledge, larger treatment effects among households without electricity access cannot be used to infer that increasing electricity access would mitigate the impact of climate change on water collection times because electricity access is correlated with many things, including wealth. To strengthen this claim, I would like to see the authors flexibly control for observable confounders using the richness of the DHS data. These controls should be specified non-linearly (e.g. using bins) to better partial out the effect of wealth and other covariates.

(b) The DHS includes (and the authors use elsewhere in the analysis) measures related to water sources. Some of these sources require electricity and some don't. The authors could further strengthen the claim that electricity access mitigates the impact of climate on water collection times by showing that electricity access only matters for households that are sourcing their water using technologies that use electricity.

11. The authors mention more than once that longer water collection times are associated with increased disease prevalence and thus claim that their results "suggest future climate change will increase the risk of child diarrhea and mortality." This is something the authors can directly test in their data. Measures of recent child illness and death are reported in the DHS. It would greatly increase the impact of the study if the authors could provide evidence that climate-driven increases in water fetching times can be linked to higher disease prevalence.

References

Burke, M., & Emerick, K. (2016). Adaptation to climate change: Evidence from US agriculture. *American Economic Journal: Economic Policy*, 8(3), 106-140.

Druckenmiller, H., & Hsiang, S. (2018). Accounting for unobservable heterogeneity in cross section using spatial first differences (No. w25177). National Bureau of Economic Research.

Hsiang, S. (2016). Climate econometrics. *Annual Review of Resource Economics*, 8, 43-75

(Remarks on code availability)

Data are not made available, so I could not run the code. The code was mostly readable.

Reviewer #3

(Remarks to the Author)

Abstract

1. In sub-Saharan Africa (SSA), over 75% of the population must spend time walking to collect water from outside their home, with women and girls bearing the majority of the time burden.

Review comment: In sub-Saharan Africa (SSA), over 75% of the population lacks on-premises water access and must spend time walking to collect water from outside their home.

2. Our findings suggest that future climate change will increase the water fetching burden in SSA.

Review comment: It is better to specify that the water fetching burden will be more realised in rural SSA. Apply this change across the manuscript where necessary, especially in the discussion section.

Main

3. The household member responsible for water collection was only reported for 15% of observations (n=150,541); of these, adult women were responsible 67% of observations, girls 10%, and men or boys 17%, with 6% of households reporting 'other'.

Review comment: Is there a reason why men were combined with boys, but women were separated from girls?

Discussion

4. Associations between precipitation and source type use were weaker. In urban areas, precipitation was associated with decreased odds of piped water usage in time periods 90 days or shorter and increased use of well water in all time periods. For rural areas, precipitation was associated only with decreased use of boreholes.

Review comment: Does the author refer to an increase or decrease in precipitation?

5. Water delivery and energy infrastructure are often rolled out separately in rural African communities, yet our results suggest that governments may want to consider investments in integrated water- energy infrastructure to enhance climate resilience and benefit human health.

Review comment: Aside this, are there other recommendations? Can you speak more about the integrated water- energy infrastructure? Are there some local initiatives undertaken in some countries which can be used as lessons for other countries? What are the implications should the issue of water fetching burden increase?

Methods

6. We utilized both DHS and Malaria Indicator Survey (MIS) datasets available for 31 countries from 104 DHS and MIS survey rounds conducted between 1990 and 2017. Overall, data were obtained from n=985,643 household surveys.

Review comment 1: In the main section of the manuscript, you have mentioned the study was conducted in 34 countries, while in the methods, 31 countries is mentioned. Kindly make the correction where necessary.

Review comment 2: What are these 34 countries in sub-Saharan Africa used in this study? A map showing these countries would be helpful, with the possibility to present the number of individuals engaged per country.

(Remarks on code availability)

Version 1:

Reviewer comments:

Reviewer #1

(Remarks to the Author)

General comment: I would like to thank the authors for addressing my original comments. The revised version of the paper is substantially improved. Please see below some remaining remarks:

1) In answer to my remark no 6, it is now evident that the authors randomly selected one household per adjacent grid cell. The selection of one household per grid cell removes important data variation and information. In doing so they also assume that the two randomly selected households are comparable. However, in the updated version, the authors noted that they estimated "SFD models using average weather and walk times per grid cell (Table S22)".

In the methodology section please discuss the limitations of selecting one household per grid cell and to address this you have used average walking times using all households per grid cell.

2) The authors should include a brief summary of the limitations of their regression analysis and data. Not all of their regression results can be interpreted as casual. For example, the regression on water source utilizations.

3) Please provide potential explanations for the results reported in figure 4B.

Minor comments

1) In the abstract

"Rural household water fetching times were more impacted by recent weather compared to all households;"

Do you mean rural households versus urban households? All households also includes rural households.

2) Line 147

"We find that precipitation, temperature, and cooling degree days affect the one-way walking time to a household's primary drinking water source."

Where does the cooling degree days come from? How is this measured?

3) Lines 180-181

"Urban households are less impacted by precipitation and more impacted by temperature than urban households"

Do you mean rural households?

(Remarks on code availability)

Reviewer #2

(Remarks to the Author)

I appreciate the authors' efforts to revise the manuscript. I find it much improved. However, my two overarching concerns remain:

1. I remain somewhat unconvinced by the empirics.

a. I am particularly concerned about the temporal misalignment between spatially differenced pairs. The authors report that the pairs come from the same survey, so are generally within a few months of each other. I want to reiterate the issue here: spatial first differences does nothing to address temporally correlated confounders. It is therefore a major concern if the pair contains two observations from different seasons. To fix ideas, imagine that one observation is from the dry season and the other is from the rainy season. There will likely be large differences in temperature and rainfall between these two observations. But there will also be differences in many other factors that affect water fetching times (for example, time spent working in agriculture) that will confound the estimated effects. Even if the majority of pairs comprise observations taken during the same month, the results could still be driven by those taken in different seasons since these pairs will have the largest amount of variation in the differenced variables. The authors could address this concern by restricting differenced pairs to surveys within the same month, or by adding country-by-year-by-month fixed effects to the panel regressions (I tried to suggest this in my first review, but the authors interpreted it as separate country, year, and month fixed effects).

b. The SFD pairs are relatively far apart in geographic space; perhaps ~50 km on average? Can the authors show a histogram of these geographic distances and discuss the implications? I have trouble following the intuition that individual households this far apart from each other are comparable along unobservables. For context, the example given in the paper that proposes SFD is to compare neighboring households along an individual street. This concern could be, at least in part, addressed by doing more to ensure the comparability of neighboring observations. For example, can the authors ensure that they are only matching rural households with other rural households, only matching within wealth quantiles, etc.

2. I still think the analysis needs additional depth to merit publication in Nature Communications. I suggested two possible analyses to improve the causal evidence for the mediating effect of electricity access. I also suggested an extension that traced out the causal pathway between climate, water fetching times, and disease prevalence. The authors elected not to pursue any of these analyses and offered no alternative analyses to further probe the mechanisms or downstream implications.

a. Controlling for observables in DHS data. The authors argue that they should not control for wealth or other observables because they are along the causal pathway between weather and water fetching times. I don't follow this. If the idea is to isolate the impact of electricity access on mediating the weather-water fetching time relationship, removing the influence of wealth (which is correlated with both water fetching times and electricity access) would strengthen the causal claim. How is wealth along the causal chain between electricity access and water fetching times? While SFD can reduce the influence of spatially-correlated unobservables, it is not a magic bullet that means other controls should not be utilized when they are readily available. Furthermore, as discussed above, the SFD pairs are imperfect in this setting, making the use of controls even more necessary.

b. Using variation on which water sources require electricity. I understand that DHS does not report which sources use electricity, but do any of the technologies either always or never require electricity? Showing heterogeneity for even these subgroups would help improve the causal link.

c. Tracing out the causal pathway between climate and disease prevalence. The authors study the effect of climate on water fetching times. Others have studied the effect of water fetching times on disease and of climate on disease, but to my knowledge, there is not prior work that links climate to disease prevalence through water fetching times. This is the exercise I suggested in my first report. It would help quantify the significance of climate-driven increases in water fetching times.

(Remarks on code availability)

Reviewer #3

(Remarks to the Author)

After reviewing the authors' responses and the revised manuscript, I confirm that the authors have addressed my concerns and recommendations. I have no reservations about the quality of the article as it is now more robust and ready for the publication.

(Remarks on code availability)

Version 2:

Reviewer comments:

Reviewer #1

(Remarks to the Author)

The authors addressed all the issues I noted. Using the available data, they conducted numerous sensitivity/robustness analyses to assess the effects of short-term weather conditions on water collecting times.

(Remarks on code availability)

Reviewer #2

(Remarks to the Author)

1. Thank you for adding county-year-month fixed effects and pointing me towards the results with grid cell averages shown in S22. These additional analyses alleviate concerns about the temporal misalignment and individual households not being comparable over such large distances. Although the coefficient estimates are similar to those obtained in the authors primary specification, I think these two model adjustments add credibility to the results and should be reflected the main specification / main text. But ultimately this is up to the authors/editor.

2. Again, it is my personal option that there could be more depth to the analysis, but this is ultimately a question for the editor.

(Remarks on code availability)

No comment

RESPONSE TO REVIEWER COMMENTS

Reviewer #1

General comment: The main objective of this study is to assess the impact of weather variability, as measured by temperature and precipitation data, on the time required to travel to gather drinking water in SSA. The research topic is contemporary and pertinent, considering the increasing weather variability and the limited coping capacity of most households in SSA. The paper has the potential to contribute to the field of climate change and vulnerability by examining one of the dimensions of household vulnerability to the effects of climate change. In the reviewer's opinion, the paper nevertheless needs to address the following key issues:

Major comments:

1. One of the paper's objectives is to "evaluate if community electricity access mitigates the relationship between weather and the water fetching time burden." However, given that access to electricity is extremely limited in sub-Saharan Africa, particularly in rural areas, the authors did not provide a compelling reason for this. This was not explicitly motivated in the main discussion section; it simply came as a surprise that they will conduct the analysis based on access to electricity.

The authors argued that access to electricity allows households to pump water from boreholes. However, this is especially true if an electric pump is the primary pump used. Is this true in most SSA counties? If this is the case, please give some supporting data/literature. If manual borehole pump is the most widely utilised technique, the case for using electricity as the primary means of mitigation the impact of weather variability on walking time to fetch water is not compelling.

We thank the reviewer for this feedback. A growing number of borewells in Africa are powered by electricity. In our data, 11% of rural households had electricity access which was sufficient variation in electricity access to explore how electricity access mitigates the relationship between weather and water fetching burden. We believe this analysis is an important component of the paper and has policy implications, as discussed below. We have added background on the links between electricity access and water access into the introduction, as follows:

"Provision of improved, on-premises water sources can be limited by lack of electricity access. In Africa, over 40% of people lack access to electricity.¹⁹ In Nigeria, a household survey found that electricity significantly influenced main drinking water source: 93% of households with access to piped water had electricity access, and public tap usage was greater in households with electricity (76%) than those without (21%), as was borehole usage (66% with access versus 34% without access).²⁰ A separate study utilizing panel data collected between 2000 and 2020 in Africa found that electricity access was associated with increased access to safely

managed drinking water.²¹ Electricity access has yet to be linked with water proximity, yet the provision of water via improved sources often necessitates electricity access at least on the community-level to enable extracting groundwater via boreholes and transporting water to households in piped systems. Throughout SSA, both electric and manual pumps are common for extracting groundwater, and future increases in electricity provision could enable more reliable access to improved water sources.^{22,23}

2. The paper lacks a coherent conceptual framework explaining how short-term weather variability is predicted to increase water fetching time in SSA. What are the primary pathways by which weather variability leads to increased water fetching time in SSA countries? Is this because of variations in the extent of improved water infrastructure development? If yes, did you control for such infrastructure development in your regression analysis? Are your results robust when you account for the proportion of households that have access to improved water sources, such as borewells?

We thank the reviewer for this comment. Short-term weather patterns can influence water availability and therefore distance to water sources, such as when hot weather causes sources to dry up and households have to walk further to access water. We have added text to explain mechanisms by which weather may impact water access (see quoted text below). We used spatial first differences as our causal inference strategy which accounts for unobserved heterogeneity within the data. We have also added a subgroup analysis motivated by the reviewer's question on if infrastructure can affect the relationship between weather and water fetching time (see Figure 4.C). Further, since there are multiple mechanisms through which recent weather can affect water access, we also conducted an analysis to understand the effect of weather variability affects water source type usage, including whether or not weather affects if households use improved versus unimproved sources (see Figure 5).

"We elected to explore the impacts of weather on walk times over various time periods ranging from 1 week to 1 year to encompass effects ranging from short term to seasonal and annual. Selection of these time periods stems from the hypothesis that households may switch drinking water sources in response to changing weather, and that the type and magnitude of these effects may differ by time period. For example, in the very short term (7 days or less), households may switch sources immediately following heavy rainfall when they view certain source types as contaminated.²⁷ Conversely, heavier rainfall in the prior month or 3 months may enable switching to surface waters or shallow wells as availability increases.³² Periods of extensive heat and limited rainfall may result in sources drying up necessitating source switching, and longer-term weather may impact availability of groundwater sources.^{12,33} We also expect that economic factors will play a role in household source selection. Consumers may prefer surface water or shallow wells over those that charge collection fees when those source types are available.³² Seasonal to annual weather can also impact crop yields and agricultural outputs, which in turn can affect households' ability to purchase water.^{34,35}"

3. What are the policy implications and priority areas based on this study? Improve access to electricity or water infrastructure to lessen the amount of walking required to fetch water. Your policy recommendation makes no mention of the role of water infrastructure development.

We agree with the reviewer that there are important policy implications from this work. We have revised the final paragraph in the discussion section to more clearly outline these implications, including investment in provision of integrated water and electricity infrastructure as follows:

“Broadly, our results highlight the importance of climate-resilient water provision in Sub-Saharan Africa. Weather events driven by climate change threaten progress in SSA towards universal access to reliable on-plot clean drinking water outlined in Sustainable Development Goal 6. We found consistent inequalities in the impacts of climate change on water fetching, which is useful for identifying priority investment areas. Rural households, especially those without electricity access, are particularly vulnerable to the effects of weather on water fetching times. We also found that effects vary by climate zones, with those living in arid and temperate regions most vulnerable to changes in weather. Electricity access mitigated the effect of weather on water fetching times in rural areas, adding to the broad range of benefits from communities gaining access to electricity.⁵⁰ Water delivery and energy infrastructure are often rolled out separately in rural African communities,⁵¹ yet our results suggest that governments may want to consider investments in integrated water-energy infrastructure to enhance climate resilience and benefit human health.”

4. It is crucial to investigate if access to improved water infrastructure can alleviate the effects of weather variability on water fetching time. Are areas with access to improved water infrastructure better equipped to handle the impact of weather variability compared to those without such infrastructure? To do this, you can include an interaction term of water infrastructure development indicators with indicators of weather variability.

We thank the reviewer for this suggestion. We have added new analysis to the paper exploring this question by conducting spatial first differences (SFD) analyses stratified by access to improved drinking water. The effects of weather on water fetching times are consistently of lower magnitude among populations with access to improved water. These results are added to Figure 4.C, and discussion of these results have been added as follows:

“To examine whether the use of improved source types resulted in increased resilience to changes in short-term weather, we conducted SFD analyses stratified by access to improved drinking water infrastructure. Improved infrastructure was defined as the following reported primary drinking water source types: borewells/boreholes, piped water, rainwater, protected springs, and protected wells. We find that households with access to improved sources are less impacted by precipitation and temperature across all time periods; in the 365 day time period, households with access to improved sources are less impacted by precipitation and temperature by an approximate factor of 2 (Fig 4C).”

5. In relation to the prior comment, when detailing the regression approach, the authors simply indicated that they accounted for the availability of electricity and other wealth-related factors. What are the wealth-related variables you have included in your model?

In the fixed effects models we conducted as a comparison to our main analyses, we did control for wealth quintile and electricity access as an interaction term with the weather variables. The wealth variable is explained in the SI, and we have added additional text to explain how these variables are calculated by DHS.

“DHS datasets include a numerical wealth index, which is calculated by DHS from other DHS survey questions such as asset ownership, construction materials used in the home, and access to and type of water and sanitation facilities. The wealth index is calculated for each country-survey individually based on the values of these survey questions, and are not comparable across surveys. Following the method by Rutstein and Staveteig, we converted the wealth index scores to a comparable scale.”

6. The regression procedure and the variables used should be clearly stated in the main text. For example, the author said that. "First, we subset data to one of the 104 country-surveys and randomly selected one data point in each grid cell." Please explain what you mean when you say you selected one data point per grid cell. Is it one data point for the weather data, or did you choose one data point from the sample households? Then, for your dependent variable, you use the average walking time to fetch water across all sample households related to each grid.

We created a combined dataset for our analysis that includes household survey responses (walk times) linked spatially and temporally with gridded weather data leading up to the survey date for each individual household. When we select one data point per grid cell, we are selecting one point (household) that is already linked with the meteorological data on the day of the survey and preceding the survey date. To clarify this, we have added text to the manuscript, as follows:

“Households were linked to weather data based on the date of the survey and the grid cell their cluster was located in. We performed simulations to randomly select pairs of households from adjacent grid cells, difference a household’s walk time and weather data across adjacent cells, and estimated the effect of weather over various time periods on household walk time with regression models.”

“We merged household survey data with the CPC and CHIRPS meteorological data to create a combined dataset of spatiotemporal data of water fetching times and meteorological data. DHS and MIS data points were aligned spatially and temporally with CPC and CHIRPS meteorological data. For each household survey data point, we extracted daily precipitation and temperature in the survey location for 0-365 days leading up to the survey date. We created

variables containing average daily maximum temperature for several time periods leading up to the survey date: 7 days, 30 days, 90 days, 180 days, and 365 days. Precipitation variables over the same time periods were created by summing daily precipitation over the prior 7, 30, 90, 180, and 365 days, and dividing by the number of weeks per time period to calculate cm/week of precipitation leading up to the survey date.”

7. With regards to the regression of water source type utilization, the authors argued that weather variability can affect water fetching times through water source switching. However, the regression method used is not clear. Is the dependent variable change in the type of water source used (e.g., from piped to borewell?) and the independent variable change in temperature and precipitation? This should be explained clearly.

We thank the reviewer for this comment. We have clarified the text to read: *“To assess the impact of weather variation on source usage, we conducted logistic regressions with source type usage as a binary dependent variable and precipitation and temperature as independent variables.”*

We also added additional text and model formulation to the methods: *“To assess the impacts of weather on source use in our dataset, we used logistic regression, controlling for country, year, and month, and clustered standard errors by DHS household clusters:*

$$Source_{k,i} = \beta_0 + \beta_1 WV_{k,i} + \beta_2 Country_i + \beta_3 Year_i + \beta_3 Month_i$$

where Source is a binary variable indicating kth type of source (e.g. improved versus unimproved, piped versus all other sources, etc.) used by household i and all other variables as described above. Results are presented as odds ratios.”

8. There is no convincing explanation provided for the results obtained when regressing water source type on weather data. For example, why do you expected to detect no impact of precipitation on the type of water source used, however you found a significant effect of temperature in rural locations.

We thank the reviewer for this comment. We found that increased temperature was associated with households being more likely to use boreholes and wells, and less likely to use piped water and surface water. These results are in line with surface water sources drying up during hotter weather, and households switching to groundwater sources that do not require grid electricity to extract (e.g. handpumps, solar powered borewells, and dug wells). A decrease in piped water usage in rural settings could be explained by reduced access to grid electricity during hot weather. Further, increased precipitation in rural areas was associated with reduced household usage of borewells, which is in line with households using closer surface water sources when they are available (consistent with our finding of shorter walk times after recent precipitation). We have added additional text of these findings to the discussion:

“One mechanism by which weather can impact water fetching time burden is through source switching. Households often use multiple drinking water sources,³³ and our results suggest that households may switch their primary drinking water source in response to seasonal weather.

Hotter weather was associated with increased usage of boreholes and shallow wells and decreased usage of piped water and surface water. These results are consistent with previously published research in rural regions that surface waters dry up seasonally or become nonfunctional during periods of high temperatures and low rainfall.^{12,24,27,33,40} Increased usage of borewells and shallow wells, but less usage of piped water supplies, could be explained by pumps moving water through piped systems being more likely to be powered by grid electricity (which is more intermittent in hot and dry periods), while water extracted from borewells is more likely to be powered by solar power or manual hand pumps. Increased precipitation was associated with decreased borehole usage in rural households, and increased use of shallow wells in urban areas. This differential response to rainfall in urban versus rural areas could be driven by differences in water infrastructure in rural versus urban areas. Taken together with our finding that increased precipitation reduces water fetching time for households, households are likely choosing to access closer sources, such as surface water in rural areas and shallow wells in urban areas, that become available during rainy periods.”

Minor comments

The article needs proofreading.

9. In the abstract the author stated that

“In sub-Saharan Africa (SSA), over 75% of the population must spend time walking to collect water from outside their home, with women and girls bearing the majority of the time burden.”

Is this a population, or a household? Not everyone in a household spends time collocating water. If so, it should be stated that "more than 75% of the population in SSA live in households where at least one member walks to collect water from outside their home."

We thank the reviewer for this feedback. We have modified the abstract in lines 16-17 to read:
“In Sub-Saharan Africa (SSA), over 75% of the population lacks on-premises water access and household members must spend time walking to collect water from outside their home.”

Reviewer #2 (Remarks to the Author):

This article empirically estimates the impact of temperature and precipitation on water fetching times in sub-Saharan Africa. Lack of access to safe drinking water is a widespread problem globally, and water collection imposes a significant burden on many of the world’s poorest households. Understanding how this burden will be affected by climate change and identifying possible mitigation measures is an important area of research. However, while I believe the topic of the study is of sufficient general interest to merit publication in Nature Communications, I have major concerns around the design and execution of the study that call into question the reliability of the findings. Further,

the paper reads as very unfinished to me – there are multiple interesting and relatively easy to implement (with their DHS data) additional analyses that would strengthen the causal claims made in the paper, shed more light on the mechanisms and consequences of increased water collection times, and increase the impact of the paper’s results.

Technical issues

I have many technical concerns with the research design and implementation. One central issue is that the research design throws out a lot of useful variation (e.g. by only using a random subset of points in estimation and by only using spatial variation when temporal variation exists). A second major problem is that the specification of climate does not follow a large literature that finds that temperature and precipitation extremes and variability generally matter far more for the human impacts of climate than averages. Specific concerns are detailed below.

We found reviewer #2’s feedback to be extremely helpful for improving our analysis and we greatly appreciate the time this reviewer took to provide constructive feedback. We have revised our main analysis based on this reviewer’s suggestions (see response to next comment) and believe the paper is overall much improved. We also explored temperature and precipitation extremes using the spatial first differences approach, as suggested by the reviewer. We conducted two analyses where we binned weather data based on the 75th and 90th percentiles, respectively, and conducted the regressions using the binary ‘weather extreme’ variable as the independent variable. These results are presented in Tables S19 and S20. We find that extreme precipitation is associated with significant decreases in walk time across all time periods considered; for example, precipitation above the 75th percentile in the last 30 days is associated with a 26 minute decrease in walk times. Extreme temperature is associated with increases in walk times across all time periods.

10. Why do the authors only select one data point per cell? This approach throws out a lot of useful information. There are at least two alternative strategies that would allow the authors to use all the data: (a) average water fetching times for all data points within a grid cell, or (b) for neighboring grid cells that each contain more than one observation, construct more than one differenced pair using independent data points. These approaches both seem far more standard and sensible to me than bootstrapping with 1000 repetitions.

We thank the reviewer for this suggestion. We agree that using the maximum number of pairs between grid cells, rather than randomly selecting just one, strengthens our analyses. We have changed our main analysis approach to this method and have presented revised results throughout the paper. Because not all data points can be used in this approach due to unequal sample sizes within adjacent grid cells, we have maintained our simulation approach to be able to leverage all of the data points within our analyses.

We also conducted a sensitivity analysis, as suggested, by averaging weather and walk times for each grid cell. These results are presented in Table S22. Point estimates are similar to those of our main analysis.

11. The spatial first differences research design was designed for cross-sectional settings, in which no temporal variation is available (Druckenmiller & Hsiang, 2019). However, the authors do have temporal variation in both climate and water fetching times in their data. Why not use this variation to improve identification? While the DHS is not a longitudinal survey, it does provide a repeated cross-section, and the authors are already collapsing the data to a standardized grid. They could therefore use temporal variation in their treatment and outcome variables collapsed to this grid. One simple way of doing so would be to calculate long differences (Burke & Emerick, 2016) between DHS surveys. Spatial first differences could then be applied to the long-differenced data. Alternatively, the authors could use a two-way fixed effects panel design.

We agree with the reviewer that this would be a useful analysis approach, yet only a subset of countries had repeated surveys that we could leverage for this type of analysis. We explored implementing this analysis, but it resulted in a loss of 50% of our grid cells as they did not contain data with repeated surveys. While we did find similar point estimates to our main analysis, we chose not to include this analysis in the manuscript because of the limited sample size.

12. It is not clear from the manuscript whether appropriate care was taken in the construction of the data to ensure the comparability of neighboring observations and the correct matching between DHS observations and climate variables.

(a) Are the pairs of neighboring observations that are differenced restricted to the same time period? The SFD approach removes all spatially correlated unobservable confounders; it does not remove temporally correlated confounders. Temporally correlated confounders likely matter a lot in this setting, as weather is highly correlated with time-varying human activities, such as time spent working in agriculture, that also affect water fetching times. Taking differences between data points recorded in different weeks would be problematic for the 7-day results. Taking differences between data points recorded in different years would be problematic for the 365-day results.

We agree with the reviewer that it's important to have temporal matching when differencing across grid cells. Household data collected through DHS/MIS are collected by cross-sectional surveys conducted during a time span of several months to 1 year, with a median collection time of approximately 5 months. In our analysis, we restricted differencing to be only between data points collected as part of the same survey collection. When we create our pairs by differencing, the mean difference between survey collection dates is approximately 30 days. We have clarified this in the manuscript methods section as follows:

“We estimated regressions using spatial first differences (SFD) (Figure 1).³⁶ We utilized the grid cells of the temperature and precipitation data to conduct the differencing and we restricted the differencing to within each DHS country survey. Differencing was restricted to within each DHS country survey to ensure data are temporally aligned. Household data collected through DHS/MIS are collected by cross-sectional surveys conducted during a time span of several months to 1 year, with a median collection time of approximately 5 months. When we create our pairs by differencing, the mean difference between survey collection dates is approximately 30 days.”

(b) DHS observations do not come with precise GSP coordinates – the coordinates are jittered up to 10km to protect privacy. This could cause an incorrect match between a DHS observation and the temperature and precipitation measures, which is important in the SFD context because the observation would be assigned the treatment of a neighboring grid cell. One easy fix is to drop all observations within 10km of the grid cell boundary in rural areas and within 2km of the grid cell boundary in urban areas (DHS jitter is 10km in rural areas, 2km in urban areas).

Thank you for this comment. We conducted a sensitivity analysis where we removed any rural points within 5km of the border (5km was chosen considering only 1% of rural data points are jittered up to 10km) and urban points within 2km of a grid cell border and repeated our analysis. This reduced our sample size significantly, from $n=974,963$ data points to $n=715,095$. The direction and magnitude of effects estimated by this sensitivity analysis were similar to our main findings; these results are presented in Table S21.

13. The authors specify the relationship between water fetching times and temperature and precipitation as linear. They conduct no robustness to the specification of this relationship. This choice is surprising given that the climate impacts literature overwhelmingly finds non-linear relationships between temperature and precipitation and human impacts (Hsiang, 2016). For example, counts of extreme heat days are far more predictive of climate damages than average temperatures, across a wide range of outcomes. And the quantity of rainfall that arrives in short, intense bursts often matters more than the average amount of precipitation. Furthermore, climate change is expected to impact climate extremes and variability just as much as averages. Best practices in climate econometrics suggest that the specification of temperature and precipitation should be non-linear. The authors could use bins or higher order polynomials and should test robustness to different specifications.

We thank the reviewer for this important comment. We tested different model specifications and found that linear regression was the best fit (largest R^2). We compared linear regression, a polynomial of order 2, and an order 3 polynomial. Across all of these analyses, we found the best performance with a linear model and chose this for our main analysis. As described above, we also explored temperature and precipitation extremes using the spatial first differences

approach. We conducted two analyses where we binned weather data based on the 75th and 90th percentiles, respectively, and conducted the regressions using the binary ‘weather extreme’ variable as the independent variable (see results presented in Tables S19 and S20).

14. The fixed effects regression used for comparison does not have granular enough fixed effects to appropriately control for unobservable confounders. I would like to see a version of this equation with country-year-month fixed effects to account for common time trends within a country and subnational spatial fixed effects to control for time-invariant differences between locations.

We thank the reviewer for this suggestion. Our fixed effects models were already conducted with country-year-month fixed effects, and we have added an additional fixed effect at the county/subnational level to our fixed effects models and replaced the prior fixed effect results with these revised estimates; see Tables S9-18 in the SI. We found very similar point estimates when compared to fixed effects without the subnational controls.

15. The authors compare differences in treatment effects between subgroups using stratified samples. This does not allow for formal testing of these differences, such as whether climate matters more for households without access to electricity. The authors should pool the observations and include interaction terms to test for heterogenous effects. In the SFD specification, the group variables uses in the interactions should also be spatially first differenced.

We agree that the fixed effects models should include rural/urban status and electricity access as an interaction term with weather rather than stratification; we have changed our fixed effects models to report these findings. We have clarified our models and added a description of the interaction terms used for the fixed effects models and updated the results in Tables S9 – S18.

For our main SFD analysis, we feel the findings of the stratified analyses are more interpretable than those of models with an interaction term, considering the binary covariates would also need to be differenced in a SFD model.

16. The reported standard errors are extremely small. This may be because the authors do not account for spatial and temporal autocorrelation in the residuals.

We thank the reviewer for this useful feedback. We have corrected our method for calculating standard errors for our simulation results. Previously we calculated standard deviations by taking the square root of the standard deviation of the model estimates across the 1000 simulations divided by the square root of $n=1000$ iterations. We have revised models throughout to assume the simulation standard deviations are equivalent to standard errors.¹

1. Harding, B., Tremblay, C. & Cousineau, D. Standard errors: A review and evaluation of standard error estimators using Monte Carlo simulations. *TQMP* **10**, 107–123 (2014).

17. The comparison of East-West and North-South SFD specifications is intended to test the robustness of the results to using different variation in data. While the estimates across these two directions consistently have the same sign, the magnitudes are quite different, falling well outside each other's 95% confidence intervals. This could reflect standard errors that are too small (see comment above), or it could be viewed as evidence that the identifying assumptions of SFD are not satisfied.

See above response.

18. The Methods section is very thin and it is not always possible to follow what the authors have done. Much more detail is required. A few examples:

- **It's hard to tell how the weather variables are constructed and defined. This is central to the interpretation of the results. For example, the authors write: "We created variables containing average daily weather for several time periods leading up to the survey date: 12 7 days, 30 days, 90 days, 180 days, and 365 days. The precipitation variables were divided by the number of weeks per time period, so that precipitation across all time periods are reported as cm of precipitation per week." Do you mean to say that you took the sum of daily precipitation over the time-period (rather than average) and then divided by the number of weeks? That would give you precipitation in cm/week, which is what I think you intend. But then in Table S1, median precipitation (measured in cm/week) is monotonically increasing in the lag period length. This suggests either that I have misinterpreted what this measure is or that there is an error in the data processing, as there is no obvious reason why average weekly precipitation should be longer for longer lag periods (unless all surveys are conducted in dry season).**

Yes, the precipitation values are total precipitation summed over the relevant time period, divided by the number of weeks in the given time period. We do find that more surveys are conducted in the dry season than in the wet season (see Fig S5), but that makes sense that the surveys avoid time periods with the heaviest yearly rainfall, as it is difficult to conduct field studies in those conditions.

We also modified the text to clarify the methods:

"We created variables containing average daily maximum temperature for several time periods leading up to the survey date: 7 days, 30 days, 90 days, 180 days, and 365 days. Precipitation variables over several time periods were created by summing daily precipitation over the prior 7, 30, 90, 180, and 365 days, and dividing by the number of weeks per time period to calculate cm/week of precipitation leading up to the survey date."

- **The estimating equations are not reported for many regressions other than the baseline**

specification. For example, the water source type use regressions (Figure 5), and the regressions that control for wealth are missing.

We thank the reviewer for this comment. We have added estimating equations for all models used in the analyses in the methods section & SI, as applicable.

Methods:

“Fixed effects regressions were estimated at the household level for each climate variable and time period combination, including fixed effects for county/subnational district, country, year, and month:

$$WT_i = \beta_0 + \beta_1 WV_{k,i} + \beta_3 Country_i + \beta_4 County_i + \beta_5 Month_i + \beta_6 Year_i$$

where WT is the one-way walk time to water source and WV_k is the k^{th} weather variable, including daily precipitation and daily maximum temperature across all time periods.”

“To assess the impacts of weather on source use in our dataset, we used logistic regression, controlling for country, year, and month, and clustering standard errors by DHS household clusters:

$$Source_{k,i} = \beta_0 + \beta_1 WV_{k,i} + \beta_3 Country_i + \beta_5 Month_i + \beta_6 Year_i$$

where Source is a binary variable indicating use of the k^{th} drinking water source and all other variables as described above. Results are presented as odds ratios.”

In the supporting information:

To determine the impacts of rural/urban status on the relationship between weather and walk times, we included an interaction term in the models:

$$WT_i = \beta_0 + \beta_1 WV_{k,i} \times RuralUrban + \beta_3 Country_i + \beta_4 County_i + \beta_5 Month_i + \beta_6 Year_i$$

Similarly, we included an interaction term between weather and electricity access status:

$$WT_i = \beta_0 + \beta_1 WV_{k,i} \times Electricity + \beta_3 Country_i + \beta_4 County_i + \beta_5 Month_i + \beta_6 Year_i$$

• The unit of observation in the fixed effect regression is never reported. Are we to assume that these regressions implemented on the same exact data frame as the SFD for comparability?

The unit of analysis is the household level for the fixed effect regressions as well as the spatial first differences models. These models were conducted on the same data frame as the spatial first differences. We have clarified this in the manuscript as follows:

“Using the same combined dataset as in the spatial first differences approach, fixed effects regressions were estimated at the household level for each climate variable and time period combination, including fixed effects for county/subnational district, country, year, and month”

Additional analyses

19. The authors argue that expanding access to electricity should be viewed as a strategy for mitigating the negative impacts of climate on water collection times. This finding would have important implications for decision-making if it was well supported by the data. However, the analysis here is quite under-developed. It could be strengthened in at least two ways using DHS data:

(a) As the authors acknowledge, larger treatment effects among households without electricity access cannot be used to infer that increasing electricity access would mitigate the impact of climate change on water collection times because electricity access is correlated with many things, including wealth. To strengthen this claim, I would like to see the authors flexibly control for observable confounders using the richness of the DHS data. These controls should be specified non-linearly (e.g. using bins) to better partial out the effect of wealth and other covariates.

We thank the reviewer for this feedback. We expect many of the suggested covariates to be along the causal pathway between weather and water fetching time and have therefore chosen not to include them in the SFD models. We have added text to the methods addressing this, as below:

“Because SFD eliminates bias due to spatially correlated variables, we opted not to include covariates in our main models that could be along the causal pathway of weather effects on water fetching time (i.e., wealth, water infrastructure, freshwater availability).”

(b) The DHS includes (and the authors use elsewhere in the analysis) measures related to water sources. Some of these sources require electricity and some don't. The authors could further strengthen the claim that electricity access mitigates the impact of climate on water collection times by showing that electricity access only matters for households that are sourcing their water using technologies that use electricity.

We thank the reviewer for this suggestion. DHS does include information on the type of water source used, and we have added a subgroup analysis by household water source usage of the effects of weather on water fetching times (see Figure 4C). Unfortunately, information on whether a source uses electricity is not available in the DHS survey. For example, we cannot differentiate between boreholes using electric pumps, handpumps, or treadle steps for water extraction. For this reason, we were unable to implement this analysis.

20. The authors mention more than once that longer water collection times are associated with increased disease prevalence and thus claim that their results “suggest future climate change will increase the risk of child diarrhea and mortality.” This is something the authors can directly test in their data. Measures of recent child illness and death are reported in the DHS. It would greatly increase the impact of the study if the authors could provide evidence that climate-driven increases in water fetching times can be linked to higher disease prevalence.

We thank the reviewer for this comment. The analysis you are describing has been conducted and published by several others, including for Rwanda by Mukabutera et al., 2016, BMC Public Health (10.1186/s12889-016-3435-9), for all of Sub-Saharan Africa by Bandyopadhyay, Kanji, & Wang, 2012, Applied Geography (10.1016/j.apgeog.2011.07.017), and more recently for all of Sub-Saharan Africa by Kemajou in 2022 in the Journal of Water & Health (10.2166/wh.2022.199). We do agree that referencing this literature can greatly strengthen our argument, and thus have added relevant text in the introduction:

“Further, climatic variability, e.g. short-term weather, has been linked with elevated child diarrhea incidence in Africa, suggesting climate change will also have impacts on child health.”²⁹⁻³¹

References

Burke, M., & Emerick, K. (2016). Adaptation to climate change: Evidence from US agriculture. American Economic Journal: Economic Policy, 8(3), 106-140.

Druckenmiller, H., & Hsiang, S. (2018). Accounting for unobservable heterogeneity in cross section using spatial first differences (No. w25177). National Bureau of Economic Research.

Hsiang, S. (2016). Climate econometrics. Annual Review of Resource Economics, 8, 43-75

Reviewer #2 (Remarks on code availability):

Data are not made available, so I could not run the code. The code was mostly readable.

*We have provided access to the data at the following repository:
https://github.com/abharv52/Water_fetching_climate*

Reviewer #3 (Remarks to the Author):

Abstract

21. In sub-Saharan Africa (SSA), over 75% of the population must spend time walking to collect water from outside their home, with women and girls bearing the majority of the time burden.

Review comment: In sub-Saharan Africa (SSA), over 75% of the population lacks on-premises water access and must spend time walking to collect water from outside their home.

We thank the reviewer for this comment. We have edited the first sentence of the abstract to read:

“In Sub-Saharan Africa (SSA), 76% of the population lacks on-premises water access,² with an estimated 20% having to walk over 30 minutes to their primary water source.³⁻⁶”

22. Our findings suggest that future climate change will increase the water fetching burden in SSA.

Review comment: It is better to specify that the water fetching burden will be more realised in rural SSA. Apply this change across the manuscript where necessary, especially in the discussion section.

Thank you for this suggestion. We have emphasized that this will be most relevant for rural SSA, where water fetching burdens are the highest, as follows: “

“We found consistent inequalities in the impacts of climate change on water fetching, which is useful for identifying priority investment areas. Rural households, especially those without electricity access, are particularly vulnerable to the effects of weather on water fetching times.”

Main

23. The household member responsible for water collection was only reported for 15% of observations (n=150,541); of these, adult women were responsible 67% of observations, girls 10%, and men or boys 17%, with 6% of households reporting ‘other’.

Review comment: Is there a reason why men were combined with boys, but women were separated from girls?

We thank the reviewer for this comment. We have disaggregated the percentages between men and boys in updated text in lines 189-191: *“The household member responsible for water collection was only reported for 15% of observations (n=150,541); of these, adult women were responsible 67% of observations, girls 10%, men 14%, and boys 4%, with 5% of households reporting ‘other’.”*

Discussion

24. Associations between precipitation and source type use were weaker. In urban areas, precipitation was associated with decreased odds of piped water usage in time periods 90 days or shorter and increased use of well water in all time periods. For rural areas, precipitation was associated only with decreased use of boreholes.

Review comment: Does the author refer to an increase or decrease in precipitation?

We have edited the text to specify that we are referring to an increase in precipitation: *“In urban areas, increased precipitation is associated with decreased odds of piped water usage in time periods 90 days or shorter and increased use of well water in all time periods. For rural areas, increased precipitation is associated only with decreased use of boreholes.”*

25. Water delivery and energy infrastructure are often rolled out separately in rural African communities,⁴¹ yet our results suggest that governments may want to consider investments in integrated water- energy infrastructure to enhance climate resilience and benefit human health.

Review comment: Aside this, are there other recommendations? Can you speak more about the integrated water- energy infrastructure? Are there some local initiatives undertaken in some countries which can be used as lessons for other countries? What are the implications should the issue of water fetching burden increase?

We have added additional text to the final paragraph of the discussion, including discussion of how our results can identify areas for priority investment and the potential for solar powered water systems, as follows:

“Our results suggest that diverse energy sources could improve climate-resilience of water supplies, as solar power would be available during hot and dry periods, while hydropower is available during rainy periods.⁵¹ Water delivery and energy infrastructure are often rolled out separately in rural African communities,⁵² yet our results suggest that governments may want to consider investments in integrated water-energy infrastructure to enhance climate resilience and benefit human health.”

Methods

26. We utilized both DHS and Malaria Indicator Survey (MIS) datasets available for 31 countries from 104 DHS and MIS survey rounds conducted between 1990 and 2017. Overall, data were obtained from n=985,643 household surveys.

Review comment 1: In the main section of the manuscript, you have mentioned the study was conducted in 34 countries, while in the methods, 31 countries is mentioned. Kindly make the correction where necessary.

We thank the reviewer for this comment, and can confirm the analysis included 31 countries. We have corrected this mistake to read 31 rather than 34 wherever mentioned.

Review comment 2: What are these 34 countries in sub-Saharan Africa used in this study? A map showing these countries would be helpful, with the possibility to present the number of individuals engaged per country.

We thank the reviewer for this suggestion. The countries included in our analysis are displayed in a map in Figure 2. We have also created a new figure, Figure 3, to include median walk times by country, and have also displayed the number of data points per country.

Response to reviewer comments

Reviewer #1

General comment: I would like to thank the authors for addressing my original comments. The revised version of the paper is substantially improved. Please see below some remaining remarks:

1) In answer to my remark no 6, it is now evident that the authors randomly selected one household per adjacent grid cell. The selection of one household per grid cell removes important data variation and information. In doing so they also assume that the two randomly selected households are comparable. However, in the updated version, the authors noted that they estimated “SFD models using average weather and walk times per grid cell (Table S22)”.

In the methodology section please discuss the limitations of selecting one household per grid cell and to address this you have used average walking times using all households per grid cell.

We were initially using one household per adjacent grid cell for our primary analysis, but, based on the previous comments of Reviewer 2, we changed our analysis to include the maximum number of pairs that avoided repeating observations for each grid cell pair: “First, we subset data to one of the 104 country-surveys and then randomly selected pairs of data from adjacent grid cells. The maximum number of available pairs that avoided repeating observations was used for each grid-cell pairing.”

However, we agree that it is important to mention the limitations of this approach, and have added the following text:

“By selecting the maximum number of pairings between grid cells and conducting 1000 replicates, we leverage the full variability of the dataset. However, not all households are included in each iteration of the model. To examine the effects of this data loss on the model outputs, we also estimated models using average weather and walk times per grid cell (see Table S22) such that all data points were included in one model.”

2) The authors should include a brief summary of the limitations of their regression analysis and data. Not all of their regression results can be interpreted as casual. For example, the regression on water source utilizations.

Thank you for this comment - this is an important clarification to make. We have amended the text to discuss these limitations. In the results, we added the following text: “However, the fixed effects results cannot be interpreted as causal as they may be impacted by unobserved confounders.”

We amended our discussion of the source use models in the discussion as follows:

“While the results of our source usage logistic models cannot be interpreted as causal, households often use multiple drinking water sources, and our results suggest that a household’s choice of primary drinking water source is associated with seasonal weather.”

We added the following text to the methods:

“The SFD method can identify causal effects with the assumption that changes in weather and unobservable confounders are not systematically correlated between adjacent grid cells.”³⁶

3) Please provide potential explanations for the results reported in figure 4B.

We have added the following text to the discussion:

“The magnitude of effect of temperature and precipitation on water fetching walk time differed by climate zone, with arid and temperate regions most susceptible to weather changes as compared to tropical regions. Arid regions are known to be vulnerable to the impacts of climate change due to limited infrastructure, services, and institutional capacity, poorly adapted crop varieties, livestock not coping with heat, and a reliance on rain-fed crops.⁴⁶ Tropical regions have ecosystems that are among the most resilient to climate change of all ecosystems,⁴⁷ which may increase the resiliency of water availability to changes in weather.”

Minor comments

1) In the abstract

“Rural household water fetching times were more impacted by recent weather compared to all households;”

Do you mean rural households versus urban households? All households also includes rural households.

Thank you this feedback. The sentence has been changed to read: “Rural household water fetching times were more impacted by recent weather compared to urban households”

2) Line 147

“We find that precipitation, temperature, and cooling degree days affect the one-way walking time to a household’s primary drinking water source.”

Where does the cooling degree days come from? How is this measured?

Thank you for catching this typo. We have removed the reference to cooling degree days.

3) Lines 180-181

“Urban households are less impacted by precipitation and more impacted by temperature than urban households”

Do you mean rural households?

Thank you for this feedback. We have corrected the sentence.

Reviewer #2 (Remarks to the Author):

I appreciate the authors' efforts to revise the manuscript. I find it much improved. However, my two overarching concerns remain:

1. I remain somewhat unconvinced by the empirics.

a. I am particularly concerned about the temporal misalignment between spatially differenced pairs. The authors report that the pairs come from the same survey, so are generally within a few months of each other. I want to reiterate the issue here: spatial first differences does nothing to address temporally correlated confounders. It is therefore a major concern if the pair contains two observations from different seasons. To fix ideas, imagine that one observation is from the dry season and the other is from the rainy season. There will likely be large differences in temperature and rainfall between these two observations. But there will also be differences in many other factors that affect water fetching times (for example, time spent working in agriculture) that will confound the estimated effects. Even if the majority of pairs comprise observations taken during the same month, the results could still be driven by those taken in different seasons since these pairs will have the largest amount of variation in the differenced variables. The authors could address this concern by restricting differenced pairs to surveys within the same month, or by adding country-by-year-by-month fixed effects to the panel regressions (I tried to suggest this in my first review, but the authors interpreted it as separate country, year, and month fixed effects).

We did not restrict differencing to points within the same month, but we did restrict differencing to the same country-survey and data points across adjacent grid cells are collected about 30 days apart on average. Based on the reviewer's suggestion, we re-ran the fixed effects models with country-by-year-by-month fixed effects. With the fixed effects model specified this way, many country-year-month combinations did not contain enough observations ($n > 10$) to allow for model convergence, so we dropped all country-year-months with fewer than 10 observations. In doing so, we were forced to drop $n = 133,579$ observations (14% of all data). Overall, the effect estimates are very similar to

our primary analysis with separate country-year-month fixed effects; see the table below. Given the data loss, we retained the original fixed effects models in the paper.

	New estimate (country-year- month)	original estimate
Temperature 7 day lag	0.50	0.53
Temperature 30 day lag	0.78	0.74
Temperature 90 day lag	0.88	0.71
Temperature 180 day lag	0.97	0.64
Temperature 365 day lag	1.1	1.1
Precipitation 7 day lag	-0.2	-0.4
Precipitation 30 day lag	-0.8	-0.8
Precipitation 90 day lag	-1.9	-1.3
Precipitation 180 day lag	-3.2	-2.2
Precipitation 365 day lag	-4.5	-4.2

b. The SFD pairs re relatively fair apart in geographic space; perhaps ~50 km on average? Can the authors show a histogram of these geographic distances and discuss the implications? I have trouble following the intuition that individual households this far apart from each other are comparable along unobservables. For context, the example given in the paper that proposes SFD is to compare neighboring households along an individual street. This concern could be, at least in part, addressed by doing more to ensure the comparability of neighboring observations. For example, can the authors ensure that they are only matching rural households with other rural households, only matching withing wealth quantiles, etc.

Thank you for this comment. The paper that proposes the spatial first differences technique primarily focuses on differencing between adjacent counties; the discussion of neighboring households along an individual street was an example of a simple implementation in one-dimensional space. Druckenmiller & Hsiang also discuss implementation in a two-dimensional gridded space, as we are doing. The authors also present an example of the effects of climate and soil on maize yields, where the unit of

observation is county-level averages.¹ We are not the first to employ this method on spatial data of this scale.²⁻⁶

Below is a histogram of the distances between individual households paired for the SFD method:

Median = 55km; standard deviation = 22km, 90th percentile = 84km

We also conducted SFD models using the average walk time & weather per grid cell, which are presented in Table S22, and find very similar point estimates for all models. The similarity of our main analyses and those using the averages per grid cell suggest that households being differenced are comparable.

2. I still think the analysis needs additional depth to merit publication in Nature Communications. I suggested two possible analyses to improve the causal evidence for the mediating effect of electricity access. I also suggested an extension that traced out the causal pathway between climate, water fetching times, and disease prevalence. The authors elected not to pursue any of these analyses and offered no alternative analyses to further probe the mechanisms or downstream implications.

We respectfully disagree with the reviewer. Our paper presents a novel causal inference analysis demonstrating that temperature and precipitation influence water fetching times in Sub-Saharan Africa with many robustness checks (including SFD with averages, SFD with max pairs, SFD with temperature and precipitation binned at extremes, and fixed effects models). We also include several sub-group analyses, including exploring effects for rural versus urban households, by usage of improved versus unimproved water

sources, by electricity access, and across climate regions. Further, we investigate the causal pathway with models of how weather affects water source type usage. Given that there are a multitude of benefits from reducing water fetching time, not just reduction in waterborne disease, we believe exploring downstream impacts is out of scope of this paper and could distract readers from the main findings.

a. Controlling for observables in DHS data. The authors argue that they should not control for wealth or other observables because they are along the causal pathway between weather and water fetching times. I don't follow this. If the idea is to isolate the impact of electricity access on mediating the weather-water fetching time relationship, removing the influence of wealth (which is correlated with both water fetching times and electricity access) would strengthen the causal claim. How is wealth along the causal chain between electricity access and water fetching times? While SFD can reduce the influence of spatially-correlated unobservables, it is not a magic bullet that means other controls should not be utilized when they are readily available. Furthermore, as discussed above, the SFD pairs are imperfect in this setting, making the use of controls even more necessary.

We believe that wealth is on the causal pathway between climate and water fetching walk times. Local weather and climate can drive income-generating activities, such as raising livestock and growing crops, which in turn can increase wealth.⁷ Wealth also influences water proximity by 1) allowing for purchase of water nearby, 2) allowing for drilling of a well in the household or community that enables water in or near the home, or 3) the hiring of someone else to collect water for the household. Because we wish to determine the total effect of weather on walk times, we did not control for wealth.

b. Using variation on which water sources require electricity. I understand that DHS does not report which sources use electricity, but do any of the technologies either always or never require electricity? Showing heterogeneity for even these subgroups would help improve the causal link.

Unfortunately water infrastructure is highly diverse across Africa and type of water source as recorded in the DHS surveys does not give definitive information about electricity requirements. For example, a piped water source could be gravity fed or require electricity for pumping; deep borewells (powered by diesel generators or electricity) and manual handpumps (no electricity) are also grouped together as the same water source type.

c. Tracing out the causal pathway between climate and disease prevalence. The authors study the effect of climate on water fetching times. Others have studied the effect of water fetching times on disease and of climate on disease, but to my knowledge, there is not prior work that links climate to disease prevalence through water fetching times. This is the exercise I suggested in my first report. It would help quantify the significance of climate-driven increases in water fetching times.

We agree that this is an important and interesting question to address in future research; however, it is beyond the scope of this manuscript. We have added the following text and citations to the discussion:

“Further, prior research has found that short-term weather impact child diarrheal disease incidence;^{29–31} future work is needed to determine how much of this effect is mediated through walk times.”

Reviewer #3 (Remarks to the Author):

After reviewing the authors' responses and the revised manuscript, I confirm that the authors have addressed my concerns and recommendations. I have no reservations about the quality of the article as it is now more robust and ready for the publication.

Thank you for these positive comments.

References

1. Druckenmiller, H. & Hsiang, S. *Accounting for Unobservable Heterogeneity in Cross Section Using Spatial First Differences*. (2019).
2. Fielding, D. Measuring the diversity dividend for community-level health and women's empowerment in Africa. *SSM - Popul. Health* **20**, 101294 (2022).
3. Lobell, D. B., Di Tommaso, S. & Burney, J. A. Globally ubiquitous negative effects of nitrogen dioxide on crop growth. *Sci. Adv.* **8**, eabm9909 (2022).
4. Taylor, C. A. & Schlenker, W. Environmental Drivers of Agricultural Productivity Growth: CO₂ Fertilization of US Field Crops. Working Paper at <https://doi.org/10.3386/w29320> (2021).
5. Fielding, D. & Regasa, D. Banking competition and financial inclusion: Evidence from Ethiopia. *World Dev.* **183**, 106733 (2024).
6. Linsenmeier, M. Temperature variability and long-run economic development. *J. Environ. Econ. Manag.* **121**, 102840 (2023).
7. Noack, F., Wunder, S., Angelsen, A. & Börner, J. Responses to Weather and Climate: A Cross-Section Analysis of Rural Incomes. SSRN Scholarly Paper at

<https://papers.ssrn.com/abstract=2688376> (2015).

Response to Reviewers' Comments

Reviewer #1 (Remarks to the Author):

The authors addressed all the issues I noted. Using the available data, they conducted numerous sensitivity/robustness analyses to assess the effects of short-term weather conditions on water collecting times.

We thank you for these comments.

Reviewer #2 (Remarks to the Author):

1. Thank you for adding county-year-month fixed effects and pointing me towards the results with grid cell averages shown in S22. These additional analyses alleviate concerns about the temporal misalignment and individual households not being comparable over such large distances. Although the coefficient estimates are similar to those obtained in the authors primary specification, I think these two model adjustments add credibility to the results and should be reflected the main specification / main text. But ultimately this is up to the authors/editor.

Thank you for these comments; we agree that the extensive sensitivity analyses conducted in this study at your suggestion strengthen this paper. The county-year-month-country fixed effects results have been added as the main fixed effects analysis. For the SFD analysis, we find that using grid cell averages is a useful sensitivity analysis, but that this approach results in a loss of variation that is retained in our current pair-selection strategy.

2. Again, it is my personal option that there could be more depth to the analysis, but this is ultimately a question for the editor.

We appreciate your feedback. We hope to follow up this work pursuing some of your suggestions as a future manuscript.

Reviewer #2 (Remarks on code availability):

No comment

Abstract

1. In sub-Saharan Africa (SSA), over 75% of the population must spend time walking to collect water from outside their home, with women and girls bearing the majority of the time burden.

R
e
v
i
e
w

the household member responsible for water collection was only reported for 15% of observations ($n=150,541$); of these, adult women were responsible 67% of observations, girls 10%, and men or boys 17%, with 6% of households reporting 'other'.

e

Review comment: Is there a reason why men were combined with boys, but women were separated from girls?

:

Discussion

4. Associations between precipitation and source type use were weaker. In urban areas, precipitation was associated with decreased odds of piped water usage in time periods 90 days or shorter and increased use of well water in all time periods. For rural areas, precipitation was associated only with decreased use of boreholes.

Review comment: Does the author refer to an increase or decrease in precipitation?

5. Water delivery and energy infrastructure are often rolled out separately in rural African communities,⁴¹ yet our results suggest that governments may want to consider investments in integrated water- energy infrastructure to enhance climate resilience and benefit human health.

Review comment: Aside this, are there other recommendations? Can you speak more about the integrated water- energy infrastructure? Are there some local initiatives undertaken in some countries which can be used as lessons for other countries? What are the implications should the issue of water fetching burden increase?

Methods

6. We utilized both DHS and Malaria Indicator Survey (MIS) datasets available for 31 countries from 104 DHS and MIS survey rounds conducted between 1990 and 2017. Overall, data were obtained from $n=985,643$ household surveys.

Review comment 1: In the main section of the manuscript, you have mentioned the study was conducted in 34 countries, while in the methods, 31 countries is mentioned. Kindly make the correction where necessary.

Review comment 2: What are these 34 countries in sub-Saharan Africa used in this study? A map showing these countries would be helpful, with the possibility to present the number of individuals engaged per country.